



# The impacts of uncertainties in emissions on aerosol data assimilation and short-term PM$_{2.5}$ predictions in CMAQ v5.2.1 over East Asia

Sojin Lee[1,2], Chul Han Song[*3], Kyung Man Han[3], Daven K. Henze[4], Kyunghwa Lee[3,5], Jinhyeok Yu[3], Jung-Hun Woo[6], Jia Jung[7], Yunsoo Choi[7], Pablo E. Saide[8], and Gregory R. Carmichael[9]

[1]The Seoul Institute, Seoul, South Korea
[2]Korea Institute of Atmospheric Prediction Systems (KIAPS), Seoul, South Korea
[3]School of Environmental Science and Engineering, Gwangju Institute of Science and Technology (GIST), Gwangju, South Korea
[4]Department of Mechanical Engineering, University of Colorado, Boulder, CO, USA
[5]Environmental Satellite Center, Climate and Air Quality Research Department, National Institute of Environmental Research (NIER), Incheon, South of Korea
[6]Department of Advanced Technology Fusion, Konkuk University, Seoul, 143-701, South Korea
[7]Department of Earth and Atmospheric Sciences, University of Houston, Houston, TX, USA
[8]Department of Atmospheric and Oceanic Sciences, Institute of the Environment and Sustainability, University of California, Los Angeles, CA, USA
[9]Center for Global and Regional Environmental Research, University of Iowa, Iowa City, IA, USA

*Correspondence to*: Chul Han Song (chsong@gist.ac.kr)

**Abstract.** For the purpose of improving PM prediction skills in East Asia, we estimated a new background error covariance matrix (BEC) for aerosol data assimilation using surface PM$_{2.5}$ observations that accounts for the uncertainties in anthropogenic emissions. In contrast to the conventional method to estimate the BEC that uses perturbations in meteorological data, this method additionally considered the perturbations using two different emission inventories. The impacts of the new BEC were then tested for the prediction of surface PM$_{2.5}$ over East Asia using Community Multi-scale Air Quality (CMAQ) initialized by three-dimensional variational method (3D-VAR). The surface PM$_{2.5}$ data measured at 154 sites in South Korea and 1,535 sites in China were assimilated every six hours during the Korea-United States Air Quality Study (KORUS-AQ) campaign period (1 May–14 June 2016). Data assimilation with our new BEC showed better agreement with the surface PM$_{2.5}$ observations than that with the conventional method. Our method also showed closer agreement with the observations in 24-hour PM$_{2.5}$ predictions with ~44 % fewer negative biases than the conventional method. We conclude that increased standard deviations, together with horizontal and vertical length scales in the new BEC, tend to improve the data assimilation and short-term predictions for the surface PM$_{2.5}$. This paper also suggests further research efforts devoted to estimating the BEC to improve PM$_{2.5}$ predictions.





# 1 Introduction

Particulate matter (PM) affects human health and climate change (Brook et al., 2010; Chung et al., 2005, 2012; Cohen et al., 2017). The impact of elevated levels of PM on human health, in particular, has become a central issue in Asia. One study

reported that $PM_{2.5}$ causes 63 premature deaths per 100,000 population in Asia (Apte et al., 2015). According to the World Health Organization (WHO), the mortality rate related to air quality in North Korea was 238 for every 100,000 people, the highest of 194 countries investigated (WHO, 2017). With regard to climate change, recent studies have revealed a significant direct and indirect aerosol effects and high PM levels on the radiation budget in Asia (Chung et al., 2010; Dong et al., 2016; Jung et al., 2019).

To investigate elevated PM conditions and their influences on health and climate in Asia, researchers frequently apply chemical transport models (CTMs). CTM predictions of $PM_{2.5}$ and related parameters can show appreciable errors, as evidenced by comparisons between CTM outputs and surface observations (Choi et al., 2019; Denby et al., 2008; Song et al., 2012), aircraft measurements (Kim et al., 2013, 2017b; Souri et al., 2018), and space-based observations from low earth orbit (LEO) and geostationary (GEO) satellite sensors (Han et al., 2013, 2015, 2019; Jung et al., 2019; Lee et al., 2016; Park et al.,

2014, Park et al., 2011). The inaccuracy of CTM simulations has been associated with uncertainties in emissions of primary air pollutants and meteorological fields as well as omissions of photochemical reactions occurring in chemical mechanisms (Han et al., 2013, 2015; Kim et al., 2017a; Song et al., 2012).

To enhance the performance of CTM simulations, a number of researchers have constrained the initial conditions (ICs) in CTM simulations and thus improved the predictions of PM concentrations (Adhikary et al., 2008; Collins et al., 2001; Feng

et al., 2018; Jiang et al., 2013; Liu et al., 2011; Saide et al., 2013, 2014; Tang et al., 2017). To constrain the ICs in PM predictions, studies typically apply data assimilation (DA) via merging the results of model simulations with observations to estimate a statistically optimized state which is determined by a balance between the relative uncertainties of the model and the observations. The uncertainties in the observations, such as surface PM observations and space-borne aerosol optical depth (AOD) are estimated by the error characteristics of their instruments or by comparisons with other observations

regarded as ground truth. In contrast, errors in the model simulations are represented by statistical parameters. As such parameters play a substantial role in the performance of the DA, properly accounting for their uncertainty is a necessary part of a robust assimilation system.

Model errors are generally represented by the background error covariance matrix (BEC) in variational DA methods, such as the three-dimensional variational method (3D-VAR), and the four-dimensional variational method (4D-VAR). Aerosol DA

studies (Benedetti and Fisher, 2007; Descombes et al., 2015; Fisher, 2003; Jiang et al., 2013; Liu et al., 2011; Pagowski et al., 2014; Pagowski and Grell, 2012; Pang et al., 2018; Saide et al., 2013, 2014; Schwartz et al., 2012; Tang et al., 2017) typically use the conventional method, referred to as the National Meteorological Center (NMC) method (Parrish and Derber, 1992; Rabier et al., 1998) to calculate the statistical parameters of the BEC. To model the statistical parameters, they use the differences between two simulations with perturbations in meteorological fields. For example, meteorological perturbations





can be obtained by taking the differences between simulations initialized at 12-hour and 24-hour predictions (referred to as "time-lagged forecasts"). Although the NMC method is a robust technique for constructing the BEC in meteorological DA studies, it is not directly applicable for representing the uncertainties of emissions in CTM DA studies since emissions are not a state variable propagated in time by the model (Pagowski et al., 2014; Pagowski and Grell, 2012). Developing suitable BEC estimation techniques that account for emission uncertainties is important not only owing to the way in which these

parameters are distinct from initial conditions in the conventional DA framework, but owing to their importance in chemical forecasts. Among the greatest sources of errors in CTM simulations (e.g., Elbern et al., 2000; Wu et al., 2008) are the uncertainties of emission inventories. A recent study developed a novel approach to estimate BEC incorporating uncertainties in anthropogenic emissions for $PM_{2.5}$ predictions (Kumar et al., 2019). The uncertainty in emissions was estimated by comparisons among a regional emission over the contiguous United States and global emissions inventories.

The study showed improved $PM_{2.5}$ predictions when the new BEC was applied in DA using satellite AOD data.

In this study, we account for the uncertainty of emissions in the estimation of the BEC for surface $PM_{2.5}$ observations and then evaluate its impact on the data reanalysis of $PM_{2.5}$ and $PM_{2.5}$ predictions over northeast Asia from 1 May to 14 June 2016, which was the period of the Korea–United States Air Quality Study (KORUS–AQ) campaign.

This manuscript is organized as follows. Section 2 explains the CTM and 3D-VAR method, including the process of the

BEC matrix and the use observations in the data assimilation and the validation study. Section 3 compares the new BEC to the conventional BEC. From the comparison, we evaluate the impact of the two BECs on surface $PM_{2.5}$ predictions against ground-based observations. Section 4 summarizes and concludes this study, and suggests future directions of study that extend this work.

## 2 Methods and Data

The 3D-VAR system for $PM_{2.5}$ predictions was developed using the Community Multi-scale Air Quality (CMAQ) model (Byun and Schere, 2006; Byun and Ching, 1999), with meteorological fields driven by the Weather Research and Forecast (WRF) model (Skamarock and Klemp, 2008). We applied two methods to determine the BEC for the 3D-VAR DA and used ground in situ observations to evaluate the impact of the developed system on the $PM_{2.5}$ predictions. The following sections discuss the CTM and MM (meteorological model) models, DA methods, and observations.

**2.1 Model Configuration**

We ran the CMAQ version 5.2.1 model, with meteorological fields driven by the WRF version 3.9.1.1 model, using National Center for Environmental Prediction (NCEP) Final (FNL) Operational Global Analysis data (National Centers For Environmental Prediction/National Weather Service/NOAA/U.S. Department Of Commerce, 2000) with time and spatial resolutions of six hours and $1° \times 1°$, respectively, in order to provide boundary and initial conditions for the WRF model

simulations. Running the CMAQ and WRF models in an offline mode, we used horizontal and vertical grid resolutions of 15



km × 15 km and 35 vertical sigma levels, respectively, with the model top at 50 hPa. We selected Statewide Air Pollution Research Center version 2007 (SAPRC07) gas-phase chemistry and AERO6 aerosol modules for this study. Other physical and chemical configurations for the WRF and CMAQ model simulations are described in Lee et al. (2016).

Figure 1 shows the model domain, which covered areas of China, the Korean Peninsula, and Japan. The observation sites in

China and Korea are highlighted in red and blue, respectively. All surface $PM_{2.5}$ observations from these sites were used in the 3D-VAR DA. In particular, we focused on two regions: South Korea and East China designated by blue solid line for validation of the model performance. East China included the Beijing, Tianjin, and Hebei areas (known as the Jing-Jin-Ji area), where severe haze events frequently take place and dramatically affected the air quality of South Korea (Choi et al., 2019; Huang et al., 2018; Lee et al., 2017; Liu et al., 2017).

It should be noted that to construct a BEC that account for uncertainties in emissions, we used two anthropogenic emission inventories. One was the Comprehensive Regional Emissions for Atmospheric Transport Experiments (CREATE) version 3.0 for the year 2015. This inventory specifically considers the economic and energy growth factors in China and Korea (Woo et al., 2012). The emissions were processed by the Sparse Matrix Operator Kernel Emissions (SMOKE) system developed by the United States Environmental Protection Agency (U.S. EPA). Details of the processes were described in

Woo et al. (2012). The CREATE emissions with a resolution of 0.1° × 0.1° were converted into a 15 km × 15 km horizontal resolution via a flux conserving interpolation technique.

The other inventory consisted of projected emissions from the Mosaic Asian anthropogenic emission inventory (MIX) for the year 2010 (Li et al., 2017). This inventory based on a mosaic approach combines available emission inventories from various countries. The components of the MIX for South Korea, China, and Japan were the Clean Air Policy Support System

(CAPSS) (Lee et al., 2011), the Multi-resolution Emission Inventory for China (MEIC), developed by Tsinghua University (http://www.meicmodel.org), and the Regional Emission Inventory in Asia (REAS) version 2.1 (Kurokawa et al., 2013), respectively.

To account for changes in emissions between 2010 and 2016 over South Korea and China, we modified the MIX inventory. Projection ratios for $SO_2$, $NO_x$, CO, VOC, and PM between 2010 and 2016 in China were based on a study by Zheng et al.

(2018). The ratios for South Korea were based on changes in emission between 2010 and 2015 (the latest available emission data for South Korea when this study was started). Information was obtained from the National Air Pollutant Emission report 2015, available at the National Air Pollutant Emission Service website (http://airemiss.nier.go.kr). Lacking information, we did not change emission data from other countries (e.g., Japan) inside the model domain. Using a flux conserving method, we converted the MIX emissions with 0.25° × 0.25° resolution into emission with a 15 km × 15 km resolution. Table 1

shows the annual emission fluxes of major air pollutants from the CREATE and MIX inventories over South Korea and China. The MIX emissions show larger $SO_2$ (+3%), NOx (+10%), and $PM_{2.5}$ (+3%) fluxes, and fewer CO (-8%) and non-methane volatile organic compound (NMVOC, -9%) emissions over China than the CREATE emissions. The differences in South Korea are relatively small, except for CO in the MIX emission inventory. The comparison between the emission





inventories is demonstrated with selected species of $SO_2$, NOx, and $PM_{2.5}$ over the domain (Fig. 2). In general, the CREATE

emission rates are lower than the MIX emission rates, particularly over the East China.

We added identical biogenic and biomass burning emission data to both anthropogenic emissions data and applied the Model of Emission of Gases and Aerosols from Nature (MEGAN) v3.0 (Guenther et al., 2006) to model biogenic emissions. To prepared input data for the MEGAN model, we used i) meteorology from the WRF model simulations, the configuration of which is described above; ii) leaf area index (LAI) data reprocessed with a 30 arc spatial resolution from eight day

averaged Moderate Resolution Imaging Spectroradiometer (MODIS) data in 2016 (Yuan et al., 2011); and iii) the green vegetable fraction (GVF) from the Visible Infrared Imaging Radiometer Suite (VIIRS) sensor.

The biomass burning emissions used in this study came from the Fire INventory from NCAR (FINN) v1.5 (Wiedinmyer et al., 2011). For the sake of reader convenience, we refer to the sum of biogenic, biomass burning, and two anthropogenic emissions datasets as the MIX and CREATE emission inventories.

**2.2 GSI 3DVAR System**

Figure 3 displays a schematic of the 24-hour $PM_{2.5}$ prediction system in this study. We employed Grid-point Statistical Interpolation (GSI) version 3.6 (Shao et al., 2016) supported by the Developmental Testbed Center NCEP Environmental Modeling Center for the 3D-VAR system. To update the aerosol initial conditions, we used surface $PM_{2.5}$ observations in South Korea and China measured at (00, 06, 12, and 18) UTC daily. The 3D-VAR algorithm optimizes assimilated fields

(also known as analysis fields) with observations by iterative processes to minimize the cost function ($J(x)$) defined below:

$$J(x) = (x - x_b)\mathbf{B^{-1}}(x - x_b) + (y - H(x))\mathbf{R^{-1}}(y - H(x)) \tag{1}$$

where, $x$ is the column of analysis fields, $x_b$ is the background state column that contains aerosol species from a previous stage of the prediction, $\mathbf{B}$ is the BEC matrix and $\mathbf{R}$ the observation error covariance matrix, respectively, $y$ is the observation columns, and H is the observation operator that maps model variables, generally referred to as control variables, onto

observation status. The method estimating the BEC is addressed in the last part of this section. The definition of $\mathbf{R}$ is described in Section. 2. 3.

We selected $PM_{2.5}$ as the control variable and calculated the background $PM_{2.5}$ using the equation below:

$$PM_{2.5,CMAQ} = f_{PM2.5,AT} \sum_i^l PM_{CMAQ,i} + f_{PM2.5,AC} \sum_j^m PM_{CMAQ,j} + f_{PM2.5,CO} \sum_k^n PM_{CMAQ,k} \tag{2}$$

where, $f_{PM2.5,AT}$, $f_{PM2.5,AC}$, and $f_{PM2.5,CO}$ are the $PM_{2.5}$ fractions of the Aitken, accumulation, and coarse modes, respectively.

These fractions denote the ratios of masses of aerosols with aerodynamic diameters of less than or equal to 2.5 μm in the total mass of aerosols. These ratios are recommended because concentrations of $PM_{2.5}$ and fine mode aerosols (i.e., Aitken plus accumulation) can significantly differ because a small tail of the coarse mode with $PM_{2.5}$ size distributions in the CMAQ model (Binkowski and Roselle, 2003; Jiang et al., 2006).

$PM_{CMAQ,i}$, $PM_{CMAQ,j}$, and $PM_{CMAQ,k}$ represent the mass concentrations of the CMAQ aerosol species belonging to the three

size modes. We selected 15 species for $PM_{CMAQ,i}$ (i.e. $l = 15$ in Eq. (2)), 58 species for $PM_{CMAQ,j}$ (i.e. $m = 58$ in Eq. (2)), and 7





species for $PM_{CMAQ,k}$ (i.e. $n = 7$ in Eq. (2)). The major species in $PM_{CMAQ,i}$ and $PM_{CMAQ,j}$ are the Aitken- and accumulation-mode sulfate, nitrate, primary and secondary organic aerosols (OAs), elementary carbon (EC), and trace elements and those in $PM_{CMAQ,k}$ are mostly sea-salt and dust aerosols. We calculated the mass contributions of each aerosol species to $PM_{2.5}$ before the assimilation and then used them to allocate horizontal and vertical increments from the 3D-VAR assimilation to

the CMAQ aerosol species.

The role of the BEC (i.e., **B** in Eq. (1)) is to determine how much observational information spreads to model grid points horizontally and vertically. The BEC comprises four successive matrices, all of which generally modelled independently (Descombes et al., 2015) as follow:

$$\mathbf{B} = \mathbf{U}_p \mathbf{S} \mathbf{U}_v \mathbf{U}_h \tag{3}$$

In Eq. (3), $\mathbf{U}_P$ represents the physical transform that defines the control variables related to the observations and their linear relationships. In this study, we selected $PM_{2.5}$ as the control variable and used Eq. (2) to calculate it. Thus, $\mathbf{U}_P$ is an identity matrix, **S** is a diagonal matrix representing model errors, and its diagonal components are the standard deviation of the control variables. $\mathbf{U}_v$ is a matrix for vertical transformation, estimated by vertical correlations of the control variable ($PM_{2.5}$ in this study). Similarly, $\mathbf{U}_h$ denotes the matrix of the horizontal transform defined by the horizontal correlation of $PM_{2.5}$. All

matrices were calculated by the generalized background error covariance matrix model version 2.0 (GEN_BE v2.0) (Descombes et al., 2015), with options of parabolic approximation for $\mathbf{U}_v$, and the fitting model by the Gaussian distribution for $\mathbf{U}_h$. Descombes et al. (2015) provides details of the equations for constructing the matrices.

Figure 4 presents a schematic of the BEC estimation and the 3D-VAR DA framework in this study. We ran four WRF-CMAQ model simulations without DA during the KORUS-AQ period (the WRF-CMAQ dashed-line box shown in Fig. 4).

Each simulation used either the CREATE or the MIX emissions along with the two meteorological fields denoted by "Met. 1" and "Met. 2" in Fig. 4 and initialized at 06 and 18 UTC daily, respectively. WRF simulations for MET.1 and MET.2 were preformed using initial and boundary conditions obtained from the FNL analysis data. As mentioned in Sect. 1, this method is referred to as "time-lagged forecasts" (Hoffman and Kalnay, 1983), used in the NMC method.

After the simulation, we processed the results by the interface module "CMAQ2GENBE," developed in this study. The

module extracted aerosol variables from the CMAQ model simulations, used Eq. (3) to calculate $PM_{2.5}$ fields, and converted the $PM_{2.5}$ data into inputs for GEN-BEv2.0. In the GEN-BEv2.0, we combined the inputs into three cases: NMC CREATE, NMC MIX, and NEW NMC. NMC CREATE (the green box in Fig. 4) used two simulations (simulation 1 and simulation 2 in the WRF-CMAQ box) that used the same CREATE emissions, but different time-lagged meteorological fields. The NMC MIX case (the blue box in Fig. 4) was the same as the NMC CREATE case, except it used MIX emission. Using all of the

simulation results to estimate the BEC parameters, the case of NEW NMC (the red box in Fig. 4) combined the uncertainties of both meteorology and emissions. This approach is similar to that of a previous study by Kumar et al. (2019) for satellite-retrieved AOD DA, who added a perturbation derived from a comparison of various global and regional inventories to anthropogenic emissions. Section 3.3 compares the statistical parameters among the BECs.





Finally, we applied 3D-VAR to the analysis fields using PM$_{2.5}$ observations and background fields via the GSI program. We selected the background fields from the first simulation (Met. 1 + CREATE emissions) at assimilation times 00, 06, 12, and 18 UTC daily and used the "BKOBS2GSI" module to convert the background fields and observations into binary format files available for the GSI. We analyzed only two DA cases with NEW NMC BEC and NMC CREATE BEC, to clarify the comparisons clear in Sect. 3.

### 2.3 PM$_{2.5}$ Observations

During the simulation period May 1 to 14 June, 2016, we acquired hourly surface PM$_{2.5}$ from the Korean public air quality data access system (http://www.airkorea.or.kr), operated by the Korea Environment Corporation and the China Urban Air Quality Real-time Release Platform (http://106.37.208.233:20035), managed by the Chinese Ministry of Ecology and Environment. All PM$_{2.5}$ were measured based on the tapered element oscillating microbalance method, the locations of which are presented in Fig. 1.

Since the observation data were collected without any information on measurement errors and biases, we applied an additional quality control process from previous studies (Jiang et al., 2013; Peng et al., 2018; Schwartz et al., 2012). First, we assumed that observations showing concentrations smaller than 6 μg m$^{-3}$ or larger than 800 μg m$^{-3}$ were unrealistic in our model domain, so we discarded them. Than we applied the "buddy test," which is typically used to eliminate data anomalies in meteorological observations as follows:

$$\left| \frac{1}{2}(O_{h-1} + O_{h+1}) - O_h \right| \leq a + bO_h \tag{4}$$

where $O_h$ is the 1-hour averaged PM$_{2.5}$ measured at time $h$ (hour), $O_{h-1}$ and $O_{h+1}$ are observations measured before and after 1 hour in the same locations, and a and b are the empirical parameters, 50 μg m$^{-3}$ and 0.15, respectively. We applied the second criterion only when $O_{h-1}$ and $O_{h+1}$ were measured. As mentioned in Sect. 2.2, in the observation error covariance matrix (**R**) in Eq. (1), the diagonal terms are defined as $(50$ μg m$^{-3} + 0.0075 \times$ PM$_{2.5})^2$ (Schwartz et al., 2012), and all the off-diagonal components are zero under the assumption that all errors in PM$_{2.5}$ measurements are mutually independent. We used quality-controlled surface PM$_{2.5}$ observations from 154 sites in South Korea and 1,524 sites in China for the DA every six hours, took measurements at all the Korean sites and Eastern Chinese sites (542 sites), and used them for validation studies.

### 3 Results and Discussion

This section discusses the impact of various BEC matrices on the 3D-VAR DA and the capability of the BEC estimation method to predict surface PM$_{2.5}$ over northeast Asia. Section 3.1 compares the parameters of the BEC calculated from the conventional NMC method and those from the NEW NMC method. To estimate the influence of the two, we validate the assimilated aerosol initial conditions against surface measurement data. Finally, we discuss the 24-hour prediction skills of surface PM$_{2.5}$ and changes in the PM$_{2.5}$ vertical profiles for each DA case.





### 3.1 BEC Parameters from the NEW NMC and Conventional NMC Methods

Figure 5 shows the vertical profiles of standard deviations (left), vertical length scales (middle), and horizontal length scales (right) from the NEW NMC BEC (red line), the NMC MIX (green line), and the NMC CREATE BEC (blue line) for $PM_{2.5}$ calculated by the model simulations sampled at 00, 06, 12, and 18 UTC daily over the model domain. The standard deviations of the three BECs are large, and they decrease with altitude. The standard deviations from the NMC CREATE BEC, the NMC MIX BEC, and the NEW NMC BEC are 2.05, 3.88, and 8.73 μg m$^{-3}$ at the surface, and 1.01, 1.76, and 4.68

μg m$^{-3}$ at 850 hPa, respectively. The NEW NMC BEC shows the largest standard deviations over the entire altitude, a finding similar to the those previous studies (Kumar et al., 2019; Pagowski et al., 2014) which indicates that we should account for the impact of the emission uncertainties when we address model errors in the DA in CTM simulations. Comparing the two NMC methods, we found remarkable differences below 850 hPa, which is about 1 km above the surface. This result coincides with the point at which the vertical profiles of the standard deviations in the BEC vary greatly around

the surface, where most $PM_{2.5}$ exists (Descombes et al., 2015; Tang et al., 2017).

Regarding the vertical length scales (shown in Fig. 5b), the NEW NMC BEC presents the largest values. The BECs from the NEW NMC, the NMC MIX, and the NMC CREATE have vertical lengths of 8.73, 5.88, and 2.05 grid points at the surface, respectively. All of the BECs present a decreasing pattern with altitude. This pattern is similar to the vertical profiles of the standard deviations in the BECs. The role of the vertical length scale defines the vertical extent of DA increments, the values

of which are generally related to the pattern of vertical transport in the CTM simulations (Descombes et al., 2015). Therefore, when the NEW NMC BEC is applied, the increase in the vertical length scale can translate into more active upward transport within the planetary boundary layer (PBL). The differences between the vertical length scales of the two NMC methods are not significant above the surface, indicating a likelihood that the conventional NMC method is limited in its ability to account for the impact of emission perturbations on vertical length scales.

The NEW NMC method also significantly changes the variation of the horizontal length scales in the BEC, as shown in Fig. 4c. At the surface, the values of the NEW NMC BEC, the NMC MIX BEC, and the NMC CREATE BEC are 119.71, 93.34, and 92.72, respectively. The values of the NEW NMC BEC are also significantly larger than those of the NMC BECs below 700 hPa. In the DA process, the horizontal length scale determines $PM_{2.5}$ increases in the horizontal spread of analysis. As most $PM_{2.5}$ exists around the surface, increases in the horizontal length scale in the NEW NMC BEC indicate farther spreads

in the horizontal adjacent grids. The profile of the horizontal length scale has two peaks in the vertical around 900 and 200 hPa in the NEW NMC. The other two profiles in the conventional NMC BECs have two peaks, one at the surface and the other at the top layer. In general, the profile of the horizontal length scale depends on the type of control variable. For example, a study conducted by Liu et al. (2011) calculated BEC statics of 14 aerosol species including sulfate, black carbon, organic carbon, sea-salt and dust. From the previous study, the horizontal length scales of the BECs including hydrophilic

black carbon, hydrophilic organic carbon, and find-mode dust species showed a similar vertical structure compared to those of our NMC BEC. In contrast, the scales of the BECs including hydrophobic black carbon, hydrophobic organic carbon, and





coarse-mode dust species had a similar vertical patterns compared to those of our NEW NMC BEC. The characteristics of the vertical and horizontal length scales, however, have not been fully explained in this study, thus requiring future investigation.

In summary, the NEW NMC BEC shows larger standard deviations and larger vertical and horizontal length scales than the other two conventional NMC BECs, particularly at low altitude. Therefore, the analysis fields determined by the $PM_{2.5}$ 3D-VAR process are closer to the observations and the increments expand farther both vertically and horizontally. The following sections further discuss the influences of these specifications in the BECs on analysis fields and prediction skills.

**3.2 Impact of the New Background Error Covariance matrix on Surface $PM_{2.5}$**

To assess the impact of 3D-VAR with various BECs on surface $PM_{2.5}$ analysis fields, we used the NEW NMC BEC and the NMC CREATE BEC. Hereafter, CTL (control), NMC, and NEW NMC runs will represent simulations without DA, simulation with DA using the NMC CREATE BEC, and simulation with DA using NEW NMC BEC, respectively. To focus on the impact of the changes in BECs, we conducted the CTL and other DA runs with the CREATE inventory, which was used for estimating NMC CREATE BEC, and identical meteorology (i.e., Met. 1), shown in Fig. 4.

Figure 6 and 7 show the performances of the CTL, the conventional NMC, and the NEW NMC runs for metrics: index of agreement (IOA) (which represents both errors and biases of analysis (Willmott, 1981)); mean bias (MB); and root mean square error (RMSE). We sampled the CMAQ $PM_{2.5}$ data from the simulations at 00, 06, 12, and 18 UTC daily between 1 May and 14 Jun 2016, and matched those to hourly model results at each location. Figures 6 and 7 focus on South Korea and East Asia, respectively, and total map of the metrics are presented in the supporting information (Fig. S1). The CTL run

showed relatively higher IOAs at the sites in South Korea (a mean value of 0.59) than those in the eastern part of China (a mean value of 0.47). The negative biases of the CTL run over South Korea (-7.38 μg m$^{-3}$) are relatively larger than those over East China (-4.14 μg m$^{-3}$).

The conventional NMC and NEW NMC produced significantly lower errors and biases for surface $PM_{2.5}$. For example, compared to the averages from the results of the CTL runs, IOA, RMSE, and MB in the NEW NMC were improved by 56%,

59%, and 85% over South Korea, and 62%, 40%, and 22% over East China, respectively. To highlight such improvements, "error reductions" (ER) as the ratio of the differences between the RMSEs calculated from the analysis (the conventional NMC or NEW NMC) and those from the CTL run to the RMSEs in the CTL run. The ERs for South Korea and East China presented in Fig. 8 indicate that the NEW NMC run produces more accurate prediction, particularly over South Korea (Fig S2 for total map of ERs). Table 2 summarizes the statistical performance metrics over South Korea and East China.

Collectively, we found that the predictions of the NEW NMC were more accurate than those of the conventional NMC. This enhanced performance of the NEW NMC also improve its short-term prediction skills of surface $PM_{2.5}$ over South Korea and East China, the details of which are discussed in the following section.





### 3.3 Impacts of New Background Error Covariance Matrix on 24-hour PM$_{2.5}$ predictions

The performance of the 24-hour PM$_{2.5}$ predictions with and without DA was compared with surface PM$_{2.5}$ observations.
Figure 9 compares the time-series of the hourly performances of the CTL, conventional NMC, and NEW NMC runs over South Korea (top row) and East China (bottom row), with IOA (first column), Pearson's correlation coefficient (second column), RMSE (third column), and MB (fourth column). Table 3 also summarizes these statistical performance metrics at 6-hour time intervals.

Both the conventional NMC and NEW NMC runs showed improved model performances. We found the most influential
time window to be from (+1H to + 6H). In the case of South Korea, the new NMC run, compared to the CTL run, improved IOA, R, RMSE, and MB by 24%, 53%, 24%, and 71 %, respectively. In the case of East China, the NEW NMC run improved IOA, R, RMSE, and MB by 26%, 107%, 20%, and 2%, respectively. Similar to the previous discussion in Sect 3.2, the NEW NMC run performed better than the conventional NMC run, showing an increased IOA (+7 %) and R (+12 %), and a decreased RMSE (-7%) and MB (-44%) over South Korea.

Interestingly we also found that the improvements of MBs in the NEW NMC run lasted longer than 24 hours while other improvements, such as those in the IOA, R, and the RMSE diminished with time. Improvements in the MBs for (+7H – +12H), (+13H – +18H), and (+19H – +24H) were 92%, 73%, and 89% over South Korea, and 83%, 93%, and 93% over East China, respectively. Improvements for the NMC run with the same time intervals were only 16%, 0%, and 0% over South Korea, and 6%, 9%, and 9% over East China.

Figures 10(a) through 10(f) display the first six-hour mean distributions of PM$_{2.5}$ predictions for the CTL run (first column), the differences between the conventional NMC and CTL runs (second column), and the differences between the NEW NMC and CTL runs (third column). The panels in the upper and bottom rows represent the results at the tenth modeling layer (~ 600 m) and the first modeling layer (~ 10 m), respectively. The differences between the DA performance of the NEW NMC and the conventional NMC runs were relate to the differences in the standard deviations and the horizontal and vertical
length scales of the BECs, as discussed in Sect. 2.2. The increased vertical PM$_{2.5}$ distributions as well as increased horizontal spreads around the observation sites at the time of the DA implementation for the NEW NMC are driven by enhanced horizontal length scales; this might play an important role in improving the prediction skills in areas where negative biases occur.

Figure 10 (g) also shows the vertical profiles over northeast Asia (blue solid line in Fig. 1) for the CTL run (black line), the
conventional NMC run (blue line), and the NEW NMC run (red line). At the surface and the tenth modeling layer, the NEW NMC run, compared to the other two runs, shows increased vertical distributions. The conventional NMC runs have a limited spread in the magnitude of its horizontal length scales. In the case of the conventional NMC runs, small increments extend upward from the surface, shown in Figs. 10(b), 10(e), and 10(g). These results show that DA runs with the new BEC (i.e., NEW NMC) produce the best 24-hour PM$_{2.5}$ predictions.



## 4 Conclusion


In this study, we developed a new method of estimating the BEC for PM$_{2.5}$ DA, accounting for the uncertainty in emissions, which has not been addressed by the conventional DA of surface PM$_{2.5}$ observations. To account for such emission uncertainties in the BEC, the current study utilized not only time-lagged meteorological fields but also two versions of independent anthropogenic emission inventories. In an assessment of the impact of the new BEC on PM$_{2.5}$ analysis fields and


the short-term prediction of surface PM$_{2.5}$ in the 3D-VAR framework, we designed and carried out three experiments: CTM model simulations without DA, with DA using the conventional NMC BEC, and with DA using the NEW NMC BECs, over the period 1 May to 14 June 2016 in East Asia.

We found that the new approach exhibited a tendency to generate substantially increased standard deviations, vertical length scales, and horizontal length scales in the BEC. Such increases occurred particularly near the surface (below 700 hPa).


Subsequently, the use of the NEW NMC BEC positively impacted both the performance of the DA and the 24-hour predictions of PM$_{2.5}$ and significantly reduced negative biases of the PM$_{2.5}$ predictions were (by as much as ~90%). Thus we conclude that the improvements were the result of the improved standard deviation and increased horizontal and vertical length scales in the NEW NMC BEC.

Based on the findings from this study, as well as the recent efforts in numerical weather prediction (NWP), we suggest two


directions of research that will contribute to the construction of a more robust and sophisticated BEC matrix. First, the statistical parameters in the BEC matrix can be further improved by optimization that accounts for the spatial and temporal characteristics of CTM background errors. The results of the CTL simulations (without DA) and the DA simulations showed that the performance of PM$_{2.5}$ model simulations over northeast China were poorer than that of simulations over South Korea. The background errors over China, specifically the standard deviation in the BEC matrix, should be larger than those over


South Korea. However, we did not address this issue in depth. Recent NWP DA studies have shown that the parameters in the BEC matrix can be tuned and optimized region-by-region while accounting for the characteristics of background errors (Choi et al., 2017; Song et al., 2018). Additional effort should be devoted to estimating a more sophisticated BEC with regional optimization. The impact of an optimized BEC on PM$_{2.5}$ predictions can then be tested.

Another topic of research would be the application of a hybrid method recently adapted in an NWP system for estimating the


BEC matrix (Kwon et al., 2018; Massart, 2018; Song et al., 2018) to the chemical DA. This method includes a weighted sum of a static BEC estimated by the conventional NMC method and a dynamic BEC estimated by daily ensemble predictions. It has been reported that the hybrid BEC method improves the NMC BEC by mimicking daily changes in background errors. In the construction of a hybrid BEC matrix in the chemical DA, perturbations in each ensemble run may be made created considering not only emission uncertainties but also other major uncertainties in CTM simulations, such as errors of aerosol


parameterization, initial and boundary conditions, and chemistry mechanisms. All of topics of research will be tested in the series of future studies.



**Code and data availability**

CMAQ v5.2.1 (https://doi.org/10.5281/zenodo.1212601) and WRF v3.9.1.1 (https://doi.org/10.5065/D6MK6B4K) models are both open source and publicly available. Source codes for CMAQ and WRF were downloaded at

https://github.com/USEPA/CMAQ and http://www2.mmm.ucar.edu/wrf/users/downloads.html, respectively. GSI v3.6 was downloaded at https://dtcenter.org/com-GSI/users/downloads/index.php. The sources of surface $PM_{2.5}$ observations were described in the manuscript. The CMAQ2GENBE and BKOBS2GSI developed in this study were written by NCL (https://doi.org/10.5065/D6WD3XH5) and can be obtained by contacting Sojin Lee (slee@si.re.kr).

**Author contributions**

SL developed the model code, performed the simulations and analyzed the results. CHS and KMH directed the experiments. DKH, PES and GRC conceptualized the main idea and supported computational resources. JJ and YC contributed to shaping the research and analysis. KL and JY helped analyze the results. JHW provided emission inventories and suggested the main idea. SL prepared the paper with contributions from all coauthors.

**Competing interests**

The authors declare that they have no conflict of interest.

**Acknowledgements**

This research was supported by the National Strategic Project–Fine particle of the National Research Foundation of Korea (NRF), funded by the Ministry of Science and ICT (MSIT), the Ministry of Environment, and the Ministry of Health and Welfare (MOHW) (NRF- 2017M3D8A1092022).

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



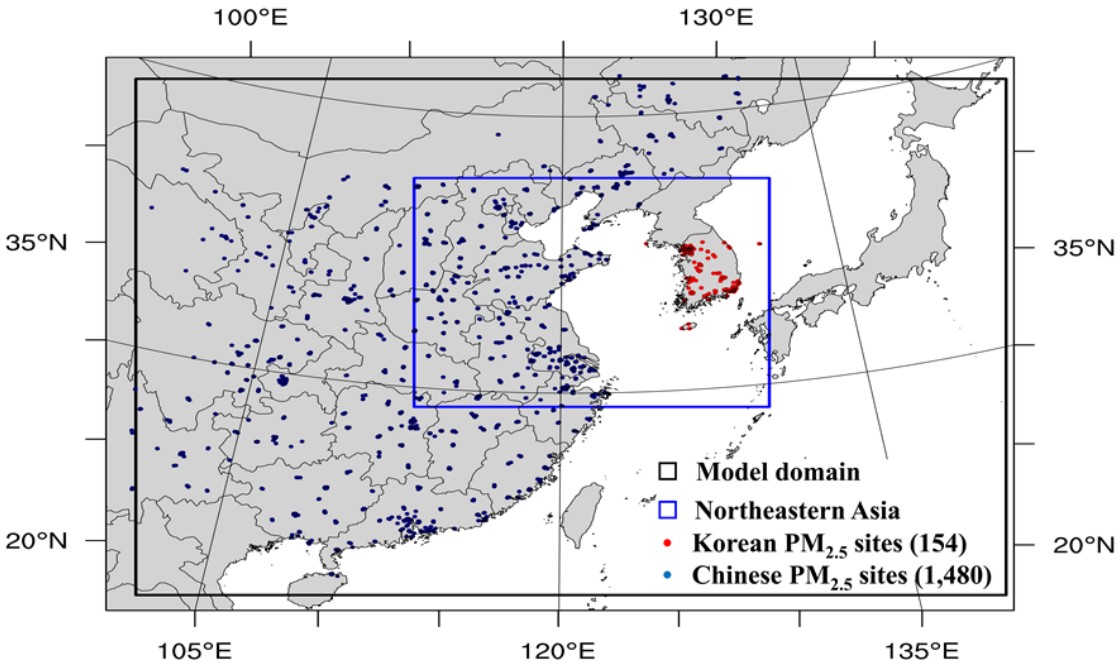

**Figure 1. Domains of CMAQ model simulations (black line) and Northeast Asia (blue line), where validation analyses were made. A total of 1,535 sites of the Chinese PM$_{2.5}$ network and 154 sites of the Korean PM$_{2.5}$ network were utilized in this study. The number of Chinese sites located in Northeast Asia is 524.**




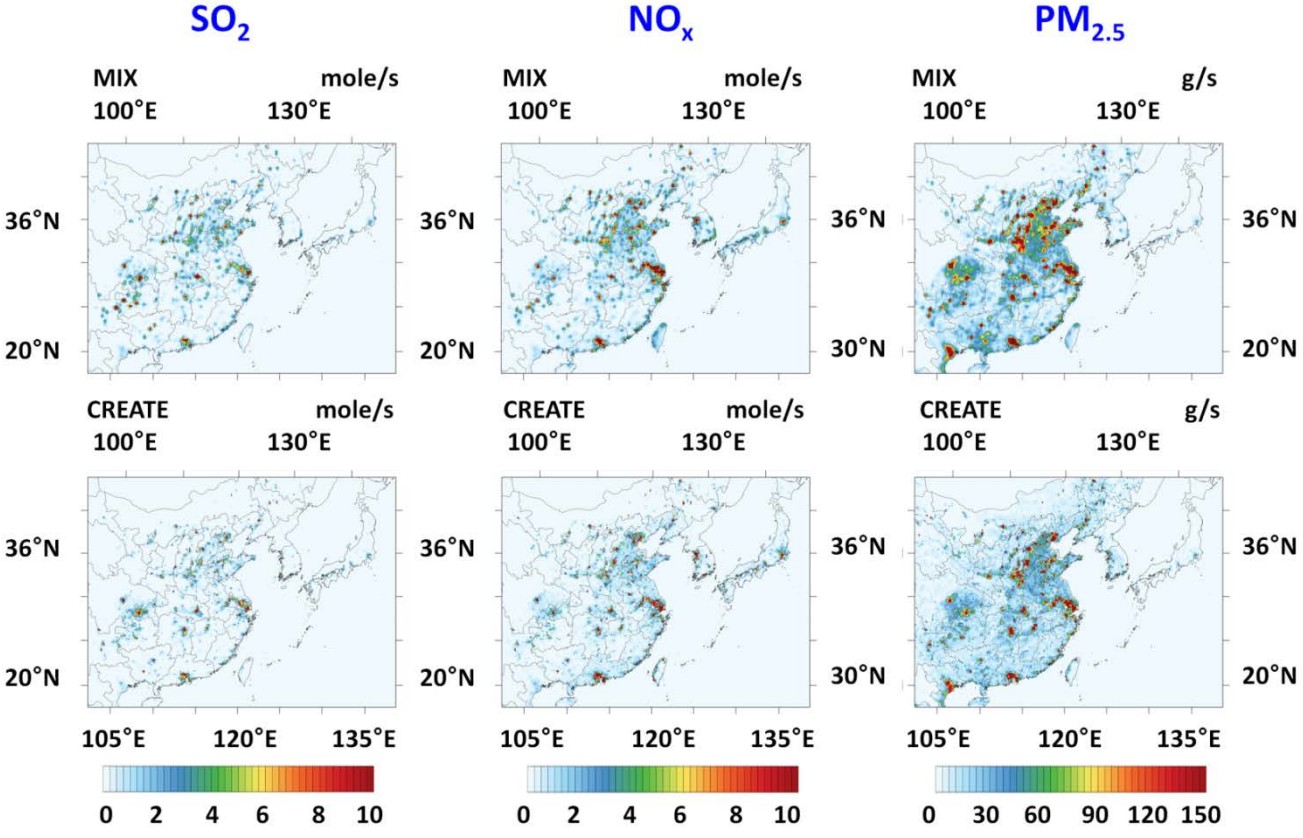

**Figure 2. Anthropogenic emission rates of SO₂ (first column), NOₓ (second column) and PM₂.₅ (third column) over the East Asia domain from MIX (upper row) and CREATE (bottom row) emission inventories.**





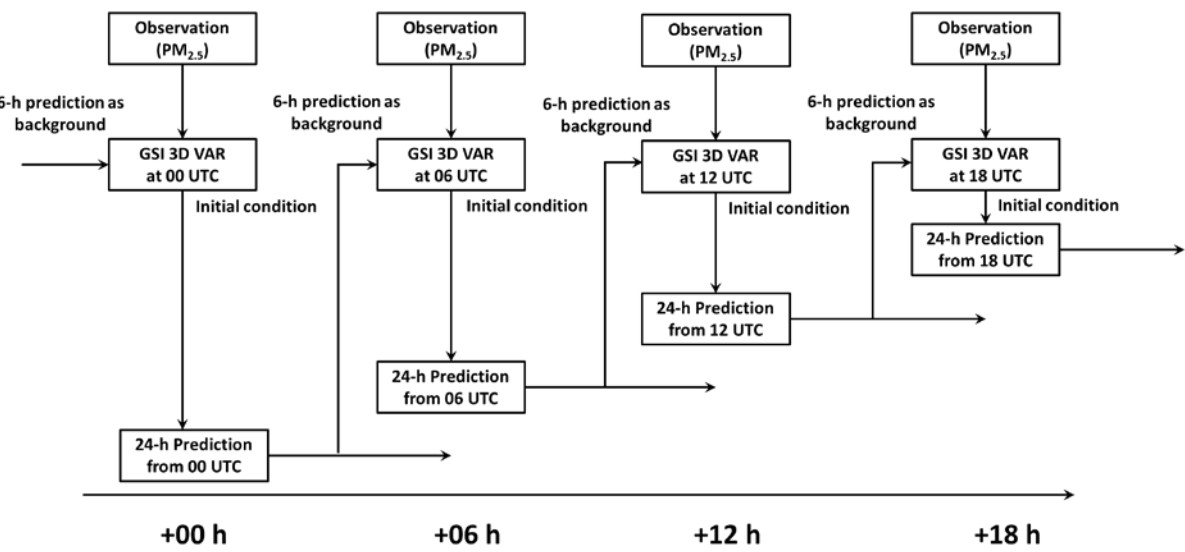


**Figure 3. Schematic of the 24-hour PM$_{2.5}$ prediction system. The 3D VAR data assimilation (DA) is applied with a 6-hour cycle using the 1-hour averaged surface PM$_{2.5}$ in South Korea and China shown in Fig. 1.**



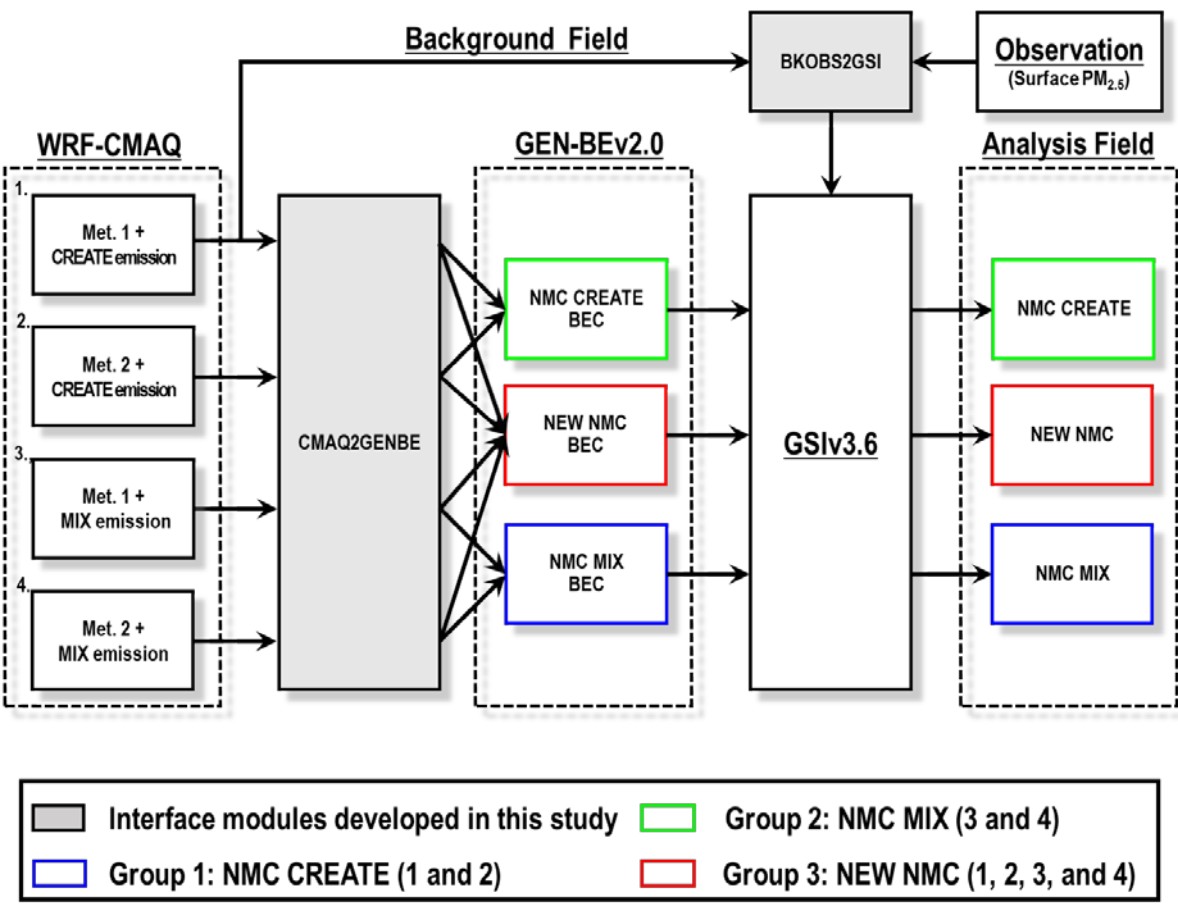

**Figure 4. Schematic of the 3D-VAR framework for surface PM$_{2.5}$. Three BECs (NMC CREATE, NMC MIX, and NEW NMC) were modeled using ~1.5 month WRF-CMAQ simulations. The two grey boxes (CMAQ2GENBE and CMAQ2GSI) are interface modules that were developed in this study. Four groups differ in the use of WRF meteorological input and the emission inventory.**





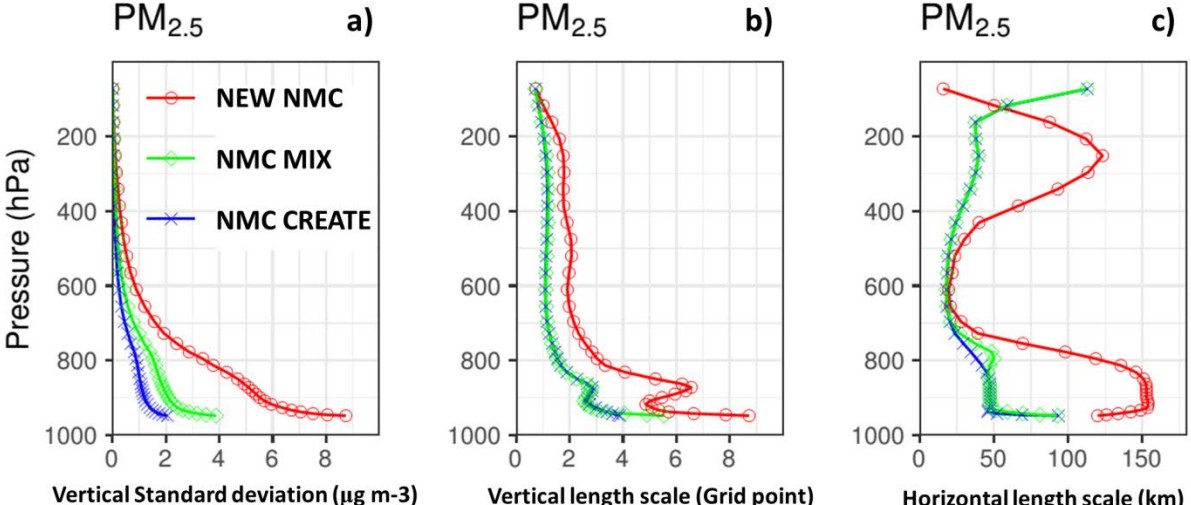

Figure 5. Vertical profiles of standard deviations (left), vertical length scales (middle), and horizontal length scales (right) for NEW NMC (red), NMC MIX (green), and NMC CREATE (blue) in the background error covariance of $PM_{2.5}$ used in this study.



**Figure 6. Statistical metrics of IOA (first column), MB (second column), and RMSE (third column) for BASE (top row), NMC (middle row), and NEW NMC (bottom row) simulations at PM$_{2.5}$ measurements over South Korea. All model data are validated at 00, 06, 12, and 18 UTC during the period of 1 May to 14 June, 2016.**



**Figure 7.** Same as Fig. 6, except for the area of East China designated by this study.






**Figure 8.** Same as Fig. 6 (top row) and Fig. 7 (bottom row), except for error reduction (ER) by the NMC (left), and the NEW NMC (right) methods.



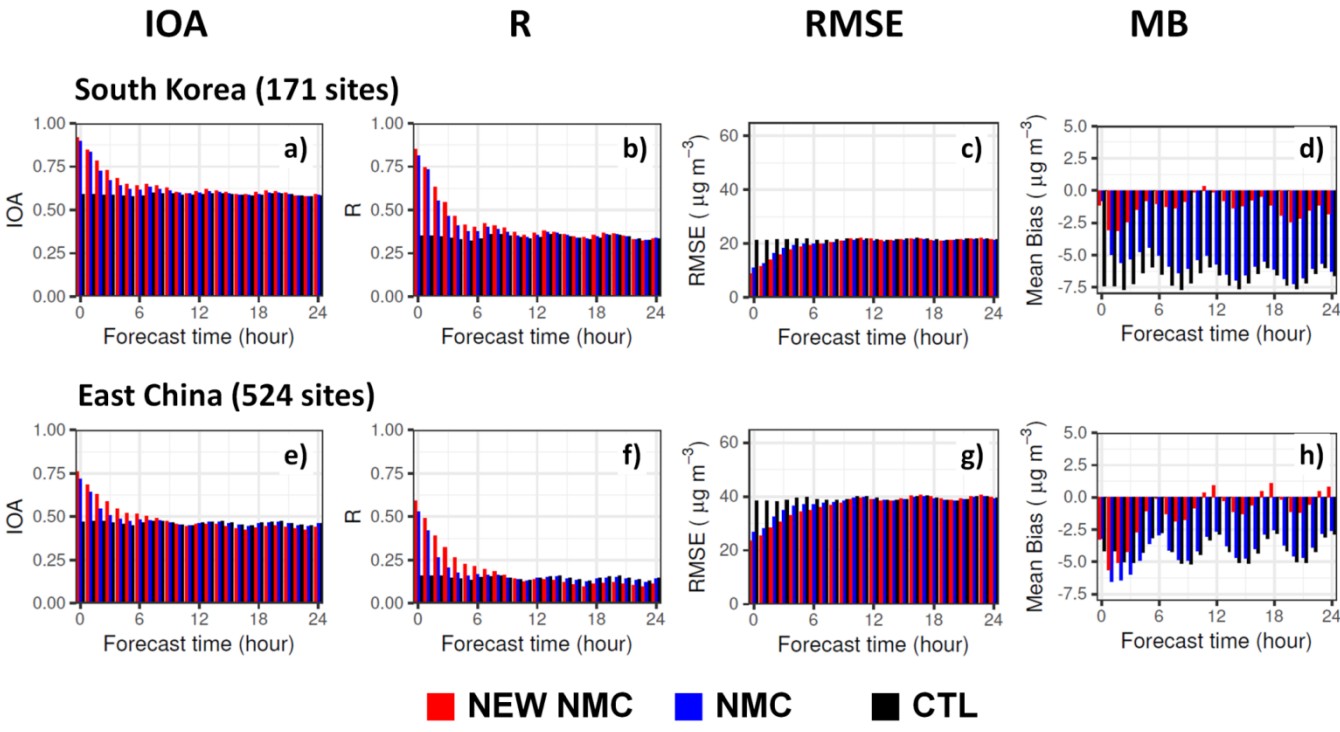

**Figure 9. Time-series plots of IOA (First column), R (second column), RMSE (third column), and MB (fourth column) for PM$_{2.5}$ predictions over South Korea (top row) and East China (bottom row) in northeast Asia.**



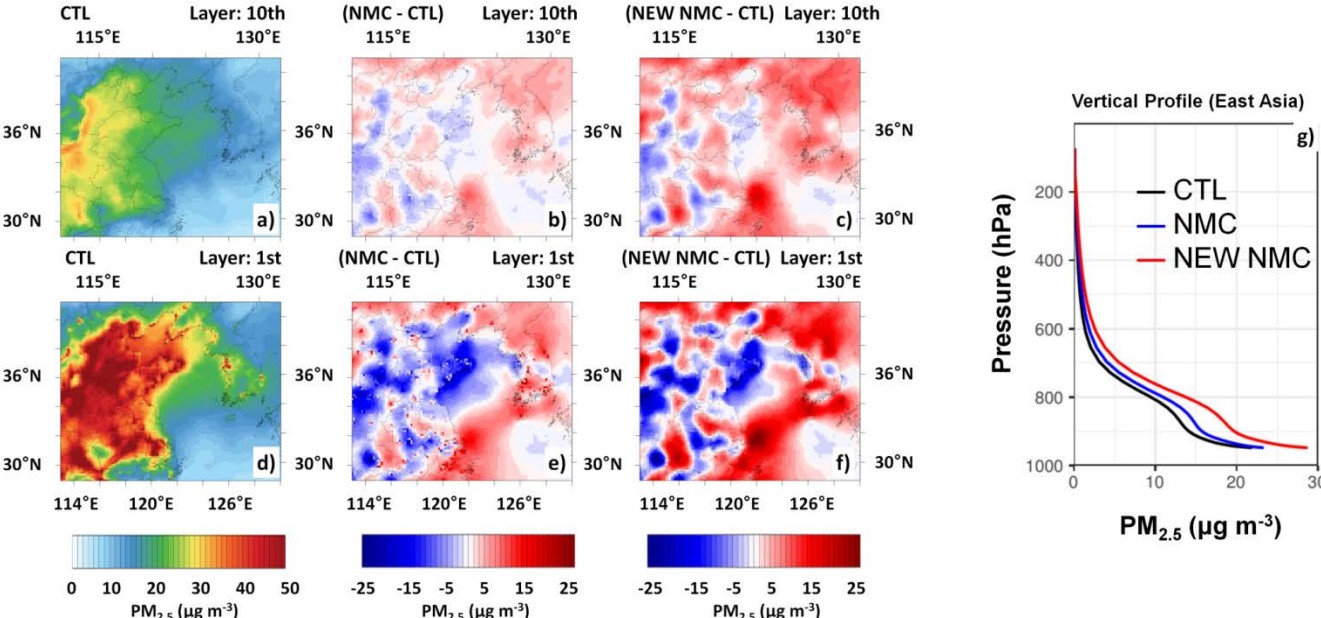

Figure 10. First 6-hour mean distributions of PM$_{2.5}$ prediction for BASE (first column), increased PM$_{2.5}$ in the NMC CREATE run (i.e. NMC CREATE – BASE, second column), and increased PM$_{2.5}$ in the NEW NMC (i.e. NEW NMC – BASE) at the 10th layer (~ 600 m, top row) and the 1st layer. Vertical profiles over northeast Asia are shown in (g) with the CTL run (black line), the NMC run (blue line), and the NEW NMC run (red line).





**Table 1. Anthropogenic emissions for selected air pollutants over South Korea and China. Here, MIX is projected from the MIX 2010 inventory (Li et al., 2017), based on the study of Zheng et al. (2016) for 2016. CREATE is based on the study of Woo et al. (2012) for the year 2015. The unit of all emissions is Gg per year.**

Unit: Gg/year

| Region | $SO_2$ | | $NO_x$ | | CO | | NMVOC | | $PM_{2.5}$ | |
|---|---|---|---|---|---|---|---|---|---|---|
| | MIX | CREATE | MIX | CREATE | MIX | CREATE | MIX | CREATE | MIX | CREATE |
| South Korea | 352 | 342 | 1,158 | 1,076 | 793 | 1,549 | 1,011 | 996 | 99 | 96 |
| China | 13,816 | 13,374 | 24,683 | 22,514 | 130,081 | 141,937 | 25,899 | 28,356 | 8,374 | 8,115 |






**Table 2. Statistical metrics from the CTL, NMC, and NEW NMC runs. The model runs were validated against surface PM$_{2.5}$ measurements at the assimilation time 00, 06, 12, and 18 UTC daily over South Korea and East China.**

| Region | Configuration | n[1] | IOA[2] | R[3] | RMSE[4] | MB[5] |
|---|---|---|---|---|---|---|
| South Korea (154 sites) | CTL | 26,650 | 0.59 | 0.35 | 21.16 | -7.38 |
| | NMC | | 0.90 | 0.81 | 10.87 | -0.78 |
| | NEW NMC | | 0.92 | 0.85 | 8.74 | -1.11 |
| East China (524 sites) | CTL | 73,948 | 0.47 | 0.16 | 38.34 | -4.14 |
| | NMC | | 0.72 | 0.53 | 26.82 | -3.18 |
| | NEW NMC | | 0.76 | 0.59 | 23.56 | -3.23 |

[1] The number of paired model and observations, [2] Index of agreement, [3] Pearson's correlation coefficient, [4] Root mean square error, and [5] Mean bias. The units of IOA and R are dimensionless. RMSE and MB have the unit µg m$^{-3}$.





**Table 3. Prediction skills of 24 hour predictions for three runs (CTL, NMC, and NEW NMC).**

| Region | Forecast hour | Configuration | N | IOA | R | RMSE | MB |
|---|---|---|---|---|---|---|---|
| South Korea (154 sites) | +1H – 6H | CTL | 152,814 | 0.58 | 0.34 | 21.49 | -6.85 |
| | | NMC | | 0.68 | 0.48 | 17.9 | -5.02 |
| | | NEW NMC | | 0.72 | 0.52 | 16.41 | -1.98 |
| | +7H – 12H | CTL | 158,154 | 0.59 | 0.35 | 21.66 | -6.86 |
| | | NMC | | 0.61 | 0.37 | 20.81 | -5.73 |
| | | NEW NMC | | 0.62 | 0.39 | 21.2 | -0.57 |
| | +13H – 18H | CTL | 156,428 | 0.59 | 0.34 | 21.69 | -6.86 |
| | | NMC | | 0.60 | 0.35 | 21.16 | -6.24 |
| | | NEW NMC | | 0.60 | 0.36 | 21.48 | -0.96 |
| | +19H – 24H | CTL | 154,647 | 0.59 | 0.34 | 21.66 | -6.88 |
| | | NMC | | 0.59 | 0.34 | 21.32 | -6.47 |
| | | NEW NMC | | 0.59 | 0.34 | 21.56 | -1.83 |
| East China (524 sites) | +1H – 6H | CTL | 436,665 | 0.46 | 0.15 | 38.93 | -4.07 |
| | | NMC | | 0.52 | 0.22 | 34.5 | -5.05 |
| | | NEW NMC | | 0.58 | 0.31 | 31.27 | -3.12 |
| | +7H – 12H | CTL | 452,033 | 0.46 | 0.15 | 39.33 | -4.18 |
| | | NMC | | 0.47 | 0.15 | 38.71 | -3.94 |
| | | NEW NMC | | 0.47 | 0.16 | 37.99 | -0.73 |
| | +13H – 18H | CTL | 446,835 | 0.46 | 0.15 | 39.36 | -4.12 |
| | | NMC | | 0.46 | 0.14 | 39.2 | -3.76 |
| | | NEW NMC | | 0.46 | 0.12 | 39.5 | -0.29 |
| | +19H – 24H | CTL | 442,031 | 0.46 | 0.14 | 39.28 | -4.06 |
| | | NMC | | 0.46 | 0.14 | 39.23 | -3.69 |
| | | NEW NMC | | 0.44 | 0.11 | 39.59 | -0.28 |