# Peer review of "Figure S1. Statistical metrics of IOA (first column), MB (second column), and RMSE (third column) for BASE (top row), NMC (middle row), and NEW NMC (bottom row) simulations at $\text{PM}_{2.5}$ measurements over Northeast Asia. All model data are validated at 00, 06, 12, and 18 UTC during the"

_Geoscientific Model Development, 2020_

## Referee Comment (RC1) · Anonymous Referee #1 · 1 Jun 2020

This manuscript described a method using two different emission inventories to estimate background error covariance (BEC) for 3D-Var chemical data assimilation, and performed the corresponding sensitivity studies compared with the traditional NMC-method generated BEC. One key issue is whether the better result achieved with the new method came from its better science or simply the larger BEC showen in Figure 5. If the difference between these two emission inventories was not so big, since they were just estimated emissions, could the new BEC still outperform the control run?

Here are the specified comments.

Line 63-75. You may need to re-write some contents. The NMC method used the laterinitialized field to reflect the more reliable prediction, or nearer the observation. Based on that difference, the model error was estimated. It is true that the NMC method may not be suitable for chemical DA, since the AQ model sometimes is more sensitive to emissions instead of the initial conditions, not because "emissions are not a state variable propagated in time" (line 67). In your later discussion (2.1 Model Configuration), there was nothing representing the observation for BEC estimation. How could you estimate the model error just based on difference between the two emission inventories?

In the main description (line 105-130) about the new BEC construction, it stated where these two emission inventories came from, but did not mentioned the uncertainty in the emissions, which was more important. In fact, once the emission uncertainty is known, one may get the BEC from perturbing a single emission inventory instead of two.

Section 2.3. This section only mentioned how to filter out bad observation, and did not tell how to estimate observation error used in the DA. Figure 5, Besides the profiles of BEC standard deviation, it is better to have a regional map showing its horizontal distributions. Figures 6-9. These comparisons have issues. It should be avoided comparing DA results to the same observations used in DA. Otherwise, the higher BEC, the better results. All the comparisons should be made after certain hours of the forward simulation to make sure that the DA will not degrade the prediction or cause side effects, e.g. RMSE would not increase.

The Figure 10 and all the statistics should follow the same comparison rule mentioned above.

---

## Referee Comment (RC2) · Anonymous Referee #2 · 15 Jun 2020

General summary This paper assesses the impacts of anthropogenic emission uncertainties on aerosol data assimilation and short-term PM2.5 predictions in East Asia. Two different anthropogenic emission inventories are used to calculate background error covariance statistical parameters. This is an interesting topic and such studies are welcome in light of the emerging needs of air quality forecasting systems especially in the developing countries. However, I have three major concerns listed below.

First, it is not clear how the authors created the New NMC background error statistical parameters. The NMC method can use only two forecasts but at line 189, the authors said they used four different forecasts to calculate the background error statistical

parameters. Please explain in detail how you did that?

Second, it is not clear how the 24 h forecasts have been performed. Are they launched everyday after assimilation at 00, 06, 12 and 18 UTC? It is difficult to follow the discussion without answering this question. I was also surprised to see that the benefits of assimilation are diminishing so rapidly. IOA, R, and RMSE go back to the CNTL experiment level in less than 12 hours. Another surprising feature is that MB in the new NMC method is larger up to 6 hours of forecast and decreases with lead time. This is unexpected because one would expect the assimilation to bring the model close to the observations at the initial time and errors would start to grow with lead time as model deficiencies start to take over. Could you please explain this?

Third, I did not find details of the independent observations used for evaluation. From the discussion, it appears that all sites were used for assimilation as well as evaluation in Figures 6 and 7. If that is the case, the results will be artificially good for the assimilation experiments.

Below please find additional specific and minor comments.

Other Specific and Minor Comments

Line 19: spell out PM. Since you are just focusing on PM2.5, you should use PM2.5 here rather than PM.

Line 28: Do you mean 44% stations showed smaller negative bias.

Line 37: Remove "a" before significant.

Line 38: I guess you meant to say "high PM levels affect the radiation budget in Asia."

Line 45: You should also highlight that inaccuracy also results from errors in initial conditions because that's what you improve with chemical data assimilation.

Line 51: Remove "the results of"

Line 126: Change "fewer" to "smaller"

Line 145: Does not GSI optimizes the background fields and creates analysis fields rather than optimizing analysis fields.

Line 148: R is

Line 150: Change "status" to "space".

Line 158: I guess you meant to say that a small tail of coarse mode distribution represents aerosols with diameter smaller than PM2.5.

Line 189: NMC method can use only two forecasts. Can you explain how you run NMC with 4 forecasts? Did you use the ensemble method of GEN_BE?

Line 194: Add "generate" before the.

Figure 4: Why are there multiple arrows pointing from CMAQ2GENBE to green, red, and blue outlined boxes?

Line 195: The authors state that they extracted background fields from the first simulation (Met. 1 + CREATE emissions) at assimilation times 00, 06, 12, and 18 UTC daily and used the "BKOBS2GSI" module to convert the background file. This is in contrast to the information shown in Figure 3 which shows that 06 hours prediction after assimilation at 00 UTC serves as initial condition for the assimilation and forecast starting at 06 UTC. This statement gives an impression that benefits of assimilation are not accumulated over time as suggested in Figure 3.

Figure 5 and Line 227: Do you use the same background error statistics at 00, 06, 12, and 18 UTC? Are the profiles shown in Figure 5 averaged over all latitude bins? What was the latitude bin in your set-up?

Line 243-244: It is also reflecting that perturbations used for NMC calculations do not account for uncertainties in vertical transport.

Line 246-247: I guess the unit of horizontal length scale is km here.

Figures 6 and 7: Looks like the color bar for RMSE is not correct. RMSE values can't be too low for the MB values shown in the middle panel. Maybe, you just copied the correlation coefficient color bar to RMSE.

Table 2: Statistical metrics in New NMC are only slightly better compared to conventional NMC. Mean Bias in New NMC is even slightly worse than conventional NMC. This indicates that emission perturbations did not have a very large impact on the analysis fields. This is somewhat unexpected considering large differences between the standard deviation between NMC and New NMC. Could you explain the reason behind this?

---

## Referee Comment (RC3) · Anonymous Referee #3 · 21 Jun 2020

This paper aims at improving the assimilation of PM2.5 observations in chemical transport model by accounting for uncertainties in emission inventories which is a major source of uncertainties for atmospheric composition forecast. The approach consists in accounting for emission uncertainties in the modelling of the background error covariance (BEC) matrix. The BEC is estimated using a modified version of the NMC statistical method which is widely used in NWP. NMC consists in approximating the background error exploiting the differences between forecasts of different lengths that verify at the same time. While in the standard NMC, the forecasts differ only from their meteorological initialization, in the modified NMC which is used in this work, emission fluxes are spatially perturbated to account for their uncertainties. This method is

applied to the Community Multi-scale Air Quality (CMAQ) model using a 3D-VAR assimilation scheme applied to PM 2.5 observations. An experiment is done over East Asia using surface PM 2.5 observations in China and South Korea. The use of the new BEC leads to more accurate PM 2.5 estimates (increase in correlation and reduced biases) assessed against independent ground observations.

This paper addresses an important issue for PM2.5 forecast. However, it is not clear what is the actual contribution of this work since the method employed to adapt the BEC was already established in Kumar et al. (2019) for satellite AOD assimilation. Most results on the impacts of the incorporation of emission uncertainties on the background error variance and the horizontal/vertical length scales have been discussed in in Kumar et al. (2019). The differences include a distinct study area, distinct emission datasets and the assimilated variable which is PM2.5 instead of AOD. While the application of the method to a distinct experiment has a potential for publication, the present paper needs further developments including a better demonstration of its scientific contributions, more accurate explanations on the methodology and a deeper analysis and discussion of the results.

General remarks ——————————

As it stands, the paper needs substantial improvements to fully understand the rationales and the key findings of this work. The scientific contribution of the paper with respect to previous studies should be better emphasized. While the overall goal is to test whether accounting for emission uncertainties in BEC modelling leads to more accurate predictions of PM 2.5, two or three discussions points should be identified in Introduction and clearly addressed in a separate section. The Introduction needs to provide more detailed and accurate background on BEC modelling and emission uncertainty quantification including a consolidated review of literature on these aspects. What are the other approaches to model BEC ? What are the limitations of the NMC method ? An alternative approach is to rely on ensemble of analyses using random perturbations of emission. Some authors have also investigated the possibility to in-

corporate the emission variables in the control vector and retrieve them using data assimilation. The use of references is not accurate enough. Some definitions (e.g. data assimilation, categories of uncertainties in CTM simulations) are not accurate enough or incorrect (e.g. background error covariance matrix). The structure of Section 2 (data and methods) needs to be improved. Some aspects of the methodology are not clear or incomplete to be fully understood and be reproductible. The paper does not provide a clear understanding of the conventional NMC versus the new NMC method. The paragraph on page 6 and Fig 4. do not provide enough details to understand how the NMC method was implemented. What are the differences between Met1 and Met2 ?. What are the actual differences between the three NMC implementations ? How the emission uncertainties have been incorporated in the NMC ? Part of the answers are in the text, but it requires too many efforts for the reader to find out.

I suggest having separate results and discussion section. I provide here some questions that could help to build the discussion:

- what are the limitations of the proposed NMC approach ?

- The variability in emission fluxes is accounted for using two inventories, is it enough to represent the spatiotemporal uncertainties in emission fluxes ? Would not be preferable to use spatial and temporal perturbations of emission fluxes based on a priori probability distribution function per type of emission.

- What are the benefits and limitations of assimilating PM2.5 compared to satellite AOD ? Is PM2.5 observation more directly related to the model variable ? What is the impact on the application of this method to over regions or historical periods which may not have PM2.5 observations.

Detailed remarks —————————-

Introduction:

- first paragraph: I suggest to give some key references for direct and indirect radiative

effects of aerosols.

- line 44: why limitation to GEO satellites ? there are a lot of atmospheric composition observations derived from LEO satellites (e.g: MODIS/TERRA, AQUA for AOD)

- "The inaccuracy of CTM simulations has been associated with uncertainties in emissions of primary air pollutants and meteorological fields as well as omissions of photochemical reactions occurring in chemical mechanisms (Han et al., 2013, 2015; Kim et al., 2017a; Song et al., 2012).": The authors provide here specific examples of source of uncertainties impacting CTM simulations. I suggest to give the main categories of source of uncertainties: drivers and forcing variables (emission inventories, meteorological fields for offline simulation, land cover . . .), model structure (e.g. photochemical reactions, more or less realistic representation of atmospheric chemistry...), model parameters.

- line 48-57: The definition of data assimilation is well known and there is no need to repeat it here. This paragraph is somehow vague. Data assimilation in NWP context has mainly two goals:provide the best estimate of initial condition and provide an estimate of the uncertainties associated with the initial state, that could include emission uncertainties.

- The list of references is sometimes too long, Authors should select two or three references for a given statement and try to be more accurate.

- line 58: The background error covariance matrix is a key component of both sequential and variational methods.

- observation errors: Observation errors include gross error (e.g. cloud detection for aod satellite), measurement errors, representativity, observation operator uncertainties

- the background error is different than the model error.

*/The model error is the departure between the true atmospheric state at time k and the model prediction. The model error is represented by a dedicated model error covariance matrix. In strong constraint 4dvar the model is assumed to be perfect and the model error is neglected. In Kalman filter, the model error covariance matrix needs to be specified.

*/The background error is the error associated to the short-term forecast. In some assimilation system the background can be a climatology and not an output of the model. Part of the error in the background is due to the model but it can also be generated by other sources of uncertainties such as emission inventories. When the BEC is flow-dependent or in sequential assimilation scheme, the BEC is updated at each cycle and thus it is also influenced by the observation error used in the previous analysis.

*/I suggest to give here the main role of the BEC in terms of information spreading, information smoothing and balance properties.

- BEC modelling: Most methods derive the statistics of the background error from the departure between the observation and the background (expel: Hollingsworth, A., and P. Lonnberg, 1986), or using a surrogate quantity whose error statistics can be a good approximation of the unknown background errors (such as NMC). More recent approaches rely on ensembles of analyses. I suggest to provide here more background information on existing approaches to model the BEC including their advantages and limitations.

- line 69 : "Among the greatest sources of errors in CTM simulations (e.g., Elbern et al., 2000; Wu et al., 2008) are the uncertainties of emission inventories". This sentence should be moved to the paragraph listing the sources of uncertainties affecting CTM simulations.

- Paragraph on methods to account for uncertainties in emission inventories: Not enough background is given on this central aspect of this paper. There are studies that have attempted to include the emission fluxes in the control vector and estimate them using ensemble data assimilation approach.

[Figure]

- need to clarify that PM2.5 is the output variable targeted in this work. What is the rationale for choosing PM2.5 instead of AOD ?

Section 2:

The structure of Section 2 needs substantial revision.

Section 2.1 includes several aspects that should be included in separate subsections, I suggest the following structure:

a/Study site and observations: • the second paragraph of Section 2.1 concerns the description of the study • it is not clear how the observations used for validation and data assimilation were selected • What is the vertical footprint of the measured PM2.5 ? How does it compare with the modelled value ?

b/model:

b1/model description: a short description of the CMAQ CTM is missing. Providing the version of the aerosol and chemistry module is not enough, key references are missing. I suggest to give the main characteristics of aerosol and chemistry schemes (e.g. number of species and reactions for the chemistry, list of aerosol species for the aerosol scheme) along the main characteristics of the atmospheric transport model: (which type of advection scheme is used )

b2/model configuration: it should address time and spatial resolution, coupling between WRT and CMAQ, temporal period, location, output variables.

c/Emission datasets: Can you justify why these two data sets have been selected for this work .

d/Data assimilation and BEC modelling: Since BEC modelling is a central aspect of the methodology, a dedicated section should explain how it is parametrized and how the NMC method is used to estimate the BEC parameters in this work.

e/Experiment design: This section should include the statements given from line 178 to

199. A table summarizing all the experiments/simulation could be helpful. The various cases of implementation of the NMC method need some clarifications.

f/Validation methodology

- a clear definition of PM2.5 is missing: what is the vertical footprint of PM2.5 ? What are the differences between the modelled and the observed PM2.5 ?

- Some aspects of the methodology are not clear or not accurate enough

- Section 2.1

line 123: which conserving method ? please give a reference

line 125-130: this belongs to Results and not to Methodology section. "The differences in South Korea are relatively small, except for CO in the MIX emission inventory.": Are you talking about the differences between the two databases ? I do not understand "except for CO in the MIX inventory"

line 130-136 on the use of MEGAN. Why are you using LAI from MODIS and GVF from VIIRS ? Are these variables required to drive MEGAN ? There is a possible inconsistency between LAI from MODIS and GVF from VIIRS ? Can you comment on it ?

- Section 2.2

the description of the cost function (l14-151) is a bit confusing. x is the control vector. x and xb contain the same variables (both are of the same size). x is the analysis and xb is the background. Are you also assimilating other variables which drive the chemistry or the transport model ? line 166 redundancy with Introduction line 172, not accurate definition of S: S represents the background error and its diagonal components are the standard deviation of the error of the background. What are the differences between the measured and the simulated PM2.5 ?

- Section 2.3:

line 211, eq 4: How a and b values have been chosen? line 213: replace 'second criterion' by 'Eq 4 criterion' line 213-215: this belong to the data assimilation section/BEC description.  c Some parts of the methodology are lacking such as the selection of observations for data assimilation versus validation.

Section 3

Section 3 should be dedicated to the presentation of the results. A separate section should address the discussions points. I shortly review the results but further review of them should be done if the manuscript is considered for publication.

- line 221-222: "To estimate the influence of the two . . ." : this belongs to the previous section

- why incorporating emission uncertainties in BEC should influence the vertical distribution for PM 2.5 ?

- l248: "In the DA process, the horizontal length scale determines PM2.5 increases in the horizontal spread of analysis" I do not understand this statement. The horizontal length scale refers to the horizontal correlation of PM2.5

- line 258: "The characteristics of the vertical and horizontal length scales, however, have not been fully explained in this study, thus requiring future": The authors should further discuss this aspect and provide possible explanations.

- Section 3.2 first Paragraph: This belongs to methodology and should be described in the experiment design section.

Conclusion

The last two paragraph should be developed in a separate discussion Section. Part of it should also be used as background information in Introduction. I can see also some redundant ideas from the Kumar et al, 2019 paper.

Technical, writing ——————————-

- Lack of references in several part of the papers

- The use of a large number of acronyms makes the reading somehow very difficult.

- Result description needs to be improved, some sentences are confusing.

- the style is frequently not appropriate with a lot of uncertain and long sentences: for example "We found that the new approach exhibited a tendency to generate substantially increased standard deviations" , a tendency to generate . . . , I suggest using more direct sentences.

---

## Referee Comment (RC4) · Anonymous Referee #4 · 25 Jun 2020

Review of

The impacts of uncertainties in emissions on aerosol data assimilation and short-term PM2.5 predictions in CMAQ v5.2.1 over East Asia

Overview:

The authors extent the NMC method to calculate back error statistics (i.e covariances, BEC) for a regional PM2.5 data assimilation system (3DVAR) to account also for the uncertainty of the emission. Using the new BEC in the DA system, they show a better agreement between the observations with the analysis and improved forecast skills (1day forecasts) for the period of the KORUS AQ measurement campaign (14.5-16.6.

2016)

General remarks

The NMC method, applied in the more common way, quantifies the uncertainty of the tracer transport from of the meteorological forecast error. But, the transport error is only one contribution to the model (background in the DA terminology) errors of air quality models. Including the uncertainty of the emissions in the BEC is therefore an interesting scientific objective. The represented method leads to increased background error standard deviations and increased vertical and horizontal length scales.

Given the structure of the 3DVAR cost function, any increase in BEC will lead to the analysis being closer to the observations, and further away from the model. Therefore, the reported better fit of the analysis with PM2.5 observations is consistent. However, there is the danger of statistical overfitting especially when the observation errors are potentially chosen to be too small. To convincingly demonstrate the improvement in the analysis requires a cross-validation approach: The randomly selected subset of the observations which is used for the evaluation should not be assimilated. I strongly recommend to carry out such a test to demonstrate the impact of the new BEC, especially the for the length scales, in a scientifically clean way.

Besides the BEC, the choice of the observation error standard deviation influences the match between the assimilated observations and the analysis. The authors should therefore provide more details how the representativeness error (i.e. of the station observations for the corresponding model grid box) of the observation has been considered. It seems that the presented approach only accounts for the instrument error but not the represenativeness error. It would be interesting to see if decreasing observation error SD has a similar influence on the analysis than increasing BEC. BEC and observations error need to be discussed in together as their relative differences determines the match of the observations with the analysis

While it is acknowledged that the uncertainty of the emissions should be considered in

the background error statistics, this quantification is not easy. Comparing two different emission data set, as done by the authors, will be dominated by the biases between the two data sets. The paper should provide more evidence that the documented in increase in BEC is a consistent estimate of the (unbiased) uncertainty of the emissions. The reader wants to know what the resulting uncertainty estimate of the emissions is.

Specific comments:

Title:

The title suggests that the reader will get an information about the uncertainty of the emissions. This seems not the case.

Abstract:

l 28: Please provide a quantification of the increase in the BE SD by taking the emission uncertainty into account.

l 53 please discuss here also the representativeness error with respect to your model resolution

l 130 Please comment if any additional temporal profiles (diurnal cycle, weekly cycle etc) were applied to the emissions during the simulation. The temporal variability might be a large source for the uncertainty of the emissions.

l 149 Theoretically speaking, the error in H is the representativeness error

l 154 Please explain if the PM components are also modified by the DA or only the diagnostic PM2.5 field.

l 185 Please provide more details here. PM2.5 simulations using the same emissions but different meteorological fields (i.e from different forecast lead times) can expected to be unbiased (i.e. inly a random error). However, this will not be the case if different emission data are used. How does the biases in the emissions turn into increased SD of the background errors. Did you remove these biases in the calculation of the variances and covariances as expected from the definitions of these statistical parameters.

l 214 I am not sure I understand this formulae. The term seems completely dominated by the 50 microgram/m3 constant value. Are you saying the SD of the observation errors is more or less 50 micorgram/m3 all the time ? That would be quite a lot. Please compare the observation error SD against the SD of the background error (2, 4 or 8 microgram, see Fig 5)

l 224 Please express these numbers also in percent w.r.t to typical PM2.5 values.

l 278 Please see my general comment. I think it is necessary to use independent (i.e. not used in the assimilation) observations to estimate the quality of the analysis.

L 298 Please provide more detail, how the analysis for diagnostic PM25 (formulae 2) is converted back in to the prognostic aerosol variables.

l 325 Please provide a quantitative information about the assumed or inferred estimate of emission uncertainty.

l 328 A proper cross validation is needed to avoid overfitting (see general comment) .

l 345 Please compare the SD of your BEC with the one derived with ensemble methods quoted in the literature.

––––––––––––––––––––––––––––––

---

## Author Comment (AC1) · 18 Nov 2020

**Response to reviewer 1**

We appreciate reviewer's thoughtful comments and suggestions, which are greatly helpful for us to improve our manuscript. The manuscript has been revised to accommodate the reviewer's comments and suggestions.

**General comment** This manuscript described a method using two different emission inventories to estimate background error covariance (BEC) for 3D-Var chemical data assimilation, and performed the corresponding sensitivity studies compared with the traditional NMC method generated BEC. One key issue is whether the better result achieved with the new method came from its better science or simply the larger BEC shown in Figure 5. If the difference between these two emission inventories was not so big, since they were just estimated emissions, could the new BEC still outperform the control run?

**Response** We agree with reviewer's comment that the differences between the two emission inventories determines the performances of data assimilation and short-term $PM_{2.5}$ predictions. However, please also note that the differences in the two emissions are not artificially created, but they are derived based on two independent emission inventories established upon independent emission statistics and factors in East Asia. Certainly, the small differences from the independent emissions will lead to small uncertainties in emission inventory. Please, also note that the differences in these two emissions are not very big (relatively small), being less than ~10%, as shown in Table 1. Regarding this point, please refer to pp. 5:161–pp 6:167 in the revised manuscript.

**Specified comments**

**Comment** Line 63-75. You may need to re-write some contents. The NMC method used the later initialized field to reflect the more reliable prediction, or nearer the observation. Based on that difference, the model error was estimated. It is true that the NMC method may not be suitable for chemical DA, since the AQ model sometimes is more sensitive to emissions instead of the initial conditions, not because "emissions are not a state variable propagated in time" (line 67). In your later discussion (2.1 Model Configuration), there was nothing representing the observation for BEC estimation. How could you estimate the model error just based on difference between the two emission inventories?

**Response** We have re-written the sentences. Please, see pp. 3:71 – pp. 3:73. Main focus of this study was on how to improve our short-term PM$_{2.5}$ prediction via new BEC using the emission perturbations. We think that the estimations of parameters in the BEC and model error against observations are a challenging topic and may be beyond the scope of this study.

**Comment** In the main description (line 105-130) about the new BEC construction, it stated where these two emission inventories came from, but did not mentioned the uncertainty in the emissions, which was more important. In fact, once the emission uncertainty is known, one may get the BEC from perturbing a single emission inventory instead of two.

**Response** Thank you for this opinion! Indeed, perturbations in single emission inventory can be used if we fully understand the sources of uncertainties in the emission inventory. It might be possible to estimate the uncertainty in emissions of selected chemical species having long lifetime or being measured intensively via inverse modeling techniques, but it is impossible to know the uncertainties of all chemical species in emission inventories. To address reviewer's comment, we have also added a recent study related to this issue. Please, see pp. 3:78 – pp. 3:81.

**Comment** Section 2.3. This section only mentioned how to filter out bad observation, and did not tell how to estimate observation error used in the DA. Figure 5, Besides the profiles of BEC standard deviation, it is better to have a regional map showing its horizontal distributions. Figures 6-9. These comparisons have issues. It should be avoided comparing DA results to the same observations used in DA. Otherwise, the higher BEC, the better results. All the comparisons should be made after certain hours of the forward simulation to make sure that the DA will not degrade the prediction or cause side effects, e.g. RMSE would not increase.

**Response** Please, see pp. 6:187 – pp. 6:189 describing the observations errors. We selected a method of binning as sampling all the horizontal grid points per each vertical level (i.e., an option of bin_type = 5 in GEN-BE v2.0). Therefore, we have a single vertical distribution of the BEC parameters. We have added this information into pp. 7:219 – pp. 7:221. To answer the review's comment, we carried out additional comparison of DA results with the 20% of independent observations which were taken out and were then used only for comparison

purpose. Regarding this point, please, refer to pp. 9:262–pp. 9:268 and pp. 11:333–pp. 11:340.

**Comment** The Figure 10 and all the statistics should follow the same comparison rule mentioned above.

**Response** Please, see the revised paragraphs mentioned above (pp. 9:262–pp. 9:268 and pp. 11:333–pp. 11:340).

---

## Author Comment (AC2) · 18 Nov 2020

**Response to reviewer 2**

We appreciate reviewer's thoughtful comments and suggestions, which are greatly helpful for us to improve our manuscript. The manuscript has been revised to accommodate the reviewer's comments.

**General summary** This paper assesses the impacts of anthropogenic emission uncertainties on aerosol data assimilation and short-term PM$_{2.5}$ predictions in East Asia. Two different anthropogenic emission inventories are used to calculate background error covariance statistical parameters. This is an interesting topic and such studies are welcome in light of the emerging needs of air quality forecasting systems especially in the developing countries. However, I have three major concerns listed below.

**General comment** First, it is not clear how the authors created the new NMC background error statistical parameters. The NMC method can use only two forecasts but at line 189, the authors said they used four different forecasts to calculate the background error statistical parameters. Please explain in detail how you did that?

**Response** For the calculations of the statistical parameters of the NMC method and our method in this study, the options of "NMC" and "ENS" were used, respectively, in GEN_BE v2.0. Regarding this point, please, refer to pp. 7:221–pp. 7:223 in the revised manuscript. Also, in order to further clarify this point, we have changed the term of "NEW NMC" to "NMC+EMIS" throughout the manuscript. We also modified Fig. 4. and added Appendix. A to remove such confusion.

**General comment** Second, it is not clear how the 24 h forecasts have been performed. Are they launched everyday after assimilation at 00, 06, 12 and 18 UTC? It is difficult to follow the discussion without answering this question.

**Response** The 24-hour forecasts were launched every 6-hour after assimilation at 00, 06, 12 and 18 UTC (four times per day). As far as we understand, this is a typical way that air quality prediction is conducted. By doing this, the benefits of assimilations can be accumulated, because each 24-h prediction uses the assimilated initial conditions using observations and

background file assimilated before 6-hour. Please, refer to modified Fig. 3 in the revised manuscript.

**General comment** I was also surprised to see that the benefits of assimilation are diminishing so rapidly. IOA, R, and RMSE go back to the CNTL experiment level in less than 12 hours. Another surprising feature is that MB in the new NMC method is larger up to 6 hours of forecast and decreases with lead time. This is unexpected because one would expect the assimilation to bring the model close to the observations at the initial time and errors would start to grow with lead time as model deficiencies start to take over. Could you please explain this?

**Response** One of the reasons for the rapid diminishing of the benefit may be that the performances of our base model (CTL) were not too good to keep the benefits of data assimilation, particularly in East China. Please refer to the RMSEs of CTL over East China are ~80% higher than these over South Korea (see Table 2 and Table S3). We think that if the CTL simulation could capture the spatial and temporal distributions of $PM_{2.5}$ with more accurate model inputs, e.g., better emissions and finer model spatial resolution over China, our method would show better performances.

**General comment** Third, I did not find details of the independent observations used for evaluation. From the discussion, it appears that all sites were used for assimilation as well as evaluation in Figures 6 and 7. If that is the case, the results will be artificially good for the assimilation experiments.

**Response** To answer the review's comment, we carried out an additional comparison of DA results with 20% of independent observations which were taken out and were then used only for comparison purpose. Regarding this point, please, refer to pp. 9:262–pp. 9:268 and pp. 11:333–pp. 11:340 in the revised manuscript.

**Other Specific and Minor Comments**

**Comment** Line 19: spell out PM. Since you are just focusing on PM2.5, you should use PM2.5 here rather than PM.

**Response** We have spelled out PM in that line. Please, check out pp. 1:19.

**Comment** Line 28: Do you mean 44% stations showed smaller negative bias.

**Response** Here, what we tried to mean was that the negative biases were reduced with the new method (NMC+EMS), compared with those from the NMC method. We have modified the sentence as followed: "Our method also showed closer agreement with the observations in 24-hour $PM_{2.5}$ predictions than the conventional method (in particular, with a ~44 % reduction of negative biases)".

Line 37: Remove "a" before significant.

**Response** We have corrected the sentence. Please, check out pp. 2:41.

Line 38: I guess you meant to say "high PM levels affect the radiation budget in Asia."

**Response** We have modified the sentence. Please refer to pp. 2:41–pp. 2:43.

**Comment** Line 45: You should also highlight that inaccuracy also results from errors in initial conditions because that's what you improve with chemical data assimilation.

**Response** We have modified the sentence. Please, see pp. 2:48–pp. 2:50.

**Comment** Line 51: Remove "the results of"

**Response** We removed it. Please, see pp. 2:55–pp. 2:57.

**Comment** Line 126: Change "fewer" to "smaller"

**Response** We have changed it. Please, check out pp. 5:154–pp. 5:156.

**Comment** Line 145: Does not GSI optimizes the background fields and creates analysis fields rather than optimizing analysis fields.

**Response** We have corrected the sentence. Please, see pp. 6:181–pp. 6:182.

**Comment** Line 148: R is

**Response** We have changed it. Please, check out pp. 6:185.

**Comment** Line 150: Change "status" to "space".

**Response** We have changed it. Please, check out pp. 6:186.

**Comment** Line 158: I guess you meant to say that a small tail of coarse mode distribution represents aerosols with diameter smaller than PM2.5.

**Response** Yes, we had tried to mean it. In order to further clarify the point, we corrected the sentences. Please, see pp. 7:195–pp. 7:197.

**Comment** Line 189: NMC method can use only two forecasts. Can you explain how you run NMC with 4 forecasts? Did you use the ensemble method of GEN_BE?

**Response** Yes, we used ensemble method in GEN_BE v2.0, as reviewer pointed out. Regarding this, please, refer to 7:221–pp. 8:223. In order to further clarify this, we have changed the term of "NEW NMC" to "NMC+EMIS" throughout the entire manuscript.

**Comment** Line 194: Add "generate" before the.

**Response** We have modified the sentence. Please, check out pp. 8:244.

**Comment** Figure 4: Why are there multiple arrows pointing from CMAQ2GENBE to green, red, and blue outlined boxes?

**Response** CMAQ2GENBE converts CMAQ outputs (NetCDF-format) into inputs for GEN-BE readable files (binary format). Two inputs from the CMAQ2GENBE were used for NMC BECs, and four inputs from the CMAQ2GENBE were used for NMC+EMIS. We have modified Fig. 4. and have added Appendix A to remove such confusion.

**Comment** Line 195: The authors state that they extracted background fields from the first simulation (Met. 1 + CREATE emissions) at assimilation times 00, 06, 12, and 18 UTC daily and used the "BKOBS2GSI" module to convert the background file. This is in contrast to the information shown in Figure 3 which shows that 06 hours prediction after assimilation at 00 UTC serves as initial condition for the assimilation and forecast starting at 06 UTC. This statement gives an impression that benefits of assimilation are not accumulated over time as suggested in Figure 3.

**Response** As we already discussed in a previous comment, the benefits of assimilations are accumulated, because each 24-h prediction uses the assimilated initial conditions using observations and background file that was assimilated before 6-hour. Please, see modified Fig. 3.

**Comment** Figure 5 and Line 227: Do you use the same background error statistics at 00, 06, 12, and 18 UTC? Are the profiles shown in Figure 5 averaged over all latitude bins? What was the latitude bin in your set-up?

**Response** We used the same background error statistics at 00, 06, 12, and 18 UTC. We selected a method of binning as sampling all the horizontal grids per each vertical level (i.e., an option of bin_type = 5 in GEN-BE v2.0). Please, refer to pp. 7:219–pp. 7:221.

**Comment** Line 243-244: It is also reflecting that perturbations used for NMC calculations do not account for uncertainties in vertical transport.

**Response** We have modified the sentence, based on your comment. Please, check out pp. 10:293–pp. 10:295.

**Comment** Line 246-247: I guess the unit of horizontal length scale is km here.

**Response** We have added the unit of horizontal length scale (km) into the sentence. Please, check out pp. 10:298.

**Comment** Figures 6 and 7: Looks like the color bar for RMSE is not correct. RMSE values can't be too low for the MB values shown in the middle panel. Maybe, you just copied the correlation coefficient color bar to RMSE.

**Response** Indeed, we made the mistakes. We corrected the color bar for RMSE in Fig. 6 and 7. Thank you for your corrections!

**Comment** Table 2: Statistical metrics in New NMC are only slightly better compared to conventional NMC. Mean Bias in New NMC is even slightly worse than conventional NMC. This indicates that emission perturbations did not have a very large impact on the analysis fields. This is somewhat unexpected considering large differences between the standard deviation between NMC and New NMC. Could you explain the reason behind this?

**Response** One of the reasons may be that our base model (CTL) does not capture the high variability of $PM_{2.5}$ over East China. We note that the RMSEs of CTL over East China are ~80% higher than those over South Korea (see Table 2 and Table S3). Despite the larger differences between the standard deviations between NMC and NMC+EMIS, it seems that our new method is limited to overcome the uncertainty in CTL simulations. We expect that if the CTL simulation could capture the spatial and temporal distributions of $PM_{2.5}$ with more accurate model inputs, e.g., better emissions and finer model spatial resolution over China, the NMC+EMIS simulation could show a better performance, as in the case of South Korea.

---

## Author Comment (AC3) · 18 Nov 2020

**Response to reviewer 3**

We appreciate reviewer's thoughtful comments and suggestions, which are greatly helpful for us to improve our manuscript. The manuscript has been revised to accommodate the reviewer's comments and suggestions.

**General comment** As it stands, the paper needs substantial improvements to fully understand the rationales and the key findings of this work. The scientific contribution of the paper with respect to previous studies should be better emphasized. While the overall goal is to test whether accounting for emission uncertainties in BEC modelling leads to more accurate predictions of $PM_{2.5}$, two or three discussions points should be identified in Introduction and clearly addressed in a separate section. The Introduction needs to provide more detailed and accurate background on BEC modelling and emission uncertainty quantification including a consolidated review of literature on these aspects. What are the other approaches to model BEC? What are the limitations of the NMC method? An alternative approach is to rely on ensemble of analyses using random perturbations of emission. Some authors have also investigated the possibility to incorporate the emission variables in the control vector and retrieve them using data assimilation. The use of references is not accurate enough. Some definitions (e.g. data assimilation, categories of uncertainties in CTM simulations) are not accurate enough or incorrect (e.g. background error covariance matrix). The structure of Section 2 (data and methods) needs to be improved. Some aspects of the methodology are not clear or incomplete to be fully understood and be re-productible. The paper does not provide a clear understanding of the conventional NMC versus the new NMC method. The paragraph on page 6 and Fig 4. do not provide enough details to understand how the NMC method was implemented. What are the differences between Met1 and Met2? What are the actual differences between the three NMC implementations? How the emission uncertainties have been incorporated in the NMC? Part of the answers are in the text, but it requires too many efforts for the reader to find out.

I suggest having separate results and discussion section. I provide here some questions that could help to build the discussion:
- what are the limitations of the proposed NMC approach ?
- The variability in emission fluxes is accounted for using two inventories, is it enough to represent the spatiotemporal uncertainties in emission fluxes ? Would not be preferable to use spatial and temporal perturbations of emission fluxes based on a priori probability distribution function per type of emission.

- What are the benefits and limitations of assimilating PM$_{2.5}$ compared to satellite AOD? Is PM$_{2.5}$ observation more directly related to the model variable? What is the impact on the application of this method to over regions or historical periods which may not have PM$_{2.5}$ observations.

**Response** Thank you for your comments. We tried our best to answer your comments and suggestions. Please, refer to our responses to your comments shown below.

**Comment** first paragraph: I suggest to give some key references for direct and indirect radiative effects of aerosols.

**Response** We have added some key references in the sentence. Please, check out pp. 2:41–pp. 2:43.

**Comment** line 44: why limitation to GEO satellites? there are a lot of atmospheric composition observations derived from LEO satellites (e.g: MODIS/TERRA, AQUA for AOD)

**Response** We discussed the advantages and disadvantages of LEO and GEO satellite-retrieved products. In particular, the fraction of satellite-derived AOD available has been low due to the high cloud fractions in East Asia. In contrast, PM$_{2.5}$ has been monitored by many ground stations, regardless of the presence of clouds. This is one of the main reasons that we chose PM$_{2.5}$ as control variable. Please, refer to the discussions at pp. 2:47–pp. 2:48 and pp. 3: 82–pp. 3:90 in the revised manuscript.

**Comment** "The inaccuracy of CTM simulations has been associated with uncertainties in emissions of primary air pollutants and meteorological fields as well as omissions of photochemical reactions occurring in chemical mechanisms (Han et al., 2013, 2015; Kim et al., 2017a; Song et al., 2012).": The authors provide here specific examples of source of uncertainties impacting CTM simulations. I suggest to give the main categories of source of uncertainties: drivers and forcing variables (emission inventories, meteorological fields for offline simulation, land cover : : :), model structure (e.g. photochemical reactions, more or less realistic representation of atmospheric chemistry: : :), model parameters.

**Response** We have tried to improve the paragraph. Please, see pp. 2: 48–pp. 2:53.

**Comment** line 48-57: The definition of data assimilation is well known and there is no need to repeat it here. This paragraph is somehow vague. Data assimilation in NWP context has mainly two goals: provide the best estimate of initial condition and provide an estimate of the uncertainties associated with the initial state, that could include emission uncertainties.

**Response** Here, we intended to discuss the roles of uncertainties in observations and models in the DA process, and then discuss the importance of model uncertainty, which is related to the main purpose of our study. We thought that this description could help readers to better understand our research purpose. We have therefore decided to keep this paragraph in the manuscript with several corrections to avoid ambiguity. Please, see pp. 2:57–pp. 2:62.

**Comment** The list of references is sometimes too long, Authors should select two or three references for a given statement and try to be more accurate

**Response** We have tried to select key references directly related to the discussions. Please, check out pp. 2:55.

**Comment** line 58: The background error covariance matrix is a key component of both sequential and variational methods

**Response** Thank you for your point! We have modified the sentence. Please, check out pp. 2:63–pp. 2:64.

**Comment** observation errors: Observation errors include gross error (e.g. cloud detection for AOD satellite), measurement errors, representativity, observation operator uncertainties
- the background error is different than the model error.
  The model error is the departure between the true atmospheric state at time k and the model prediction. The model error is represented by a dedicated model error covariance matrix. In

strong constraint 4dvar the model is assumed to be perfect and the model error is neglected. In Kalman filter, the model error covariance matrix needs to be specified.

The background error is the error associated to the short-term forecast. In some assimilation system the background can be a climatology and not an output of the model. Part of the error in the background is due to the model but it can also be generated by other sources of uncertainties such as emission inventories. When the BEC is flow-dependent or in sequential assimilation scheme, the BEC is updated at each cycle and thus it is also influenced by the observation error used in the previous analysis.

I suggest to give here the main role of the BEC in terms of information spreading, information smoothing and balance properties.

**Response** Based upon your suggestions, we have tried to modify the sentences. Please, refer to pp. 2:65–pp. 3:66. Also, in order to avoid the confusion, we have changed the term of "model error" to "background error" throughout the entire manuscript.

**Comment** BEC modelling: Most methods derive the statistics of the background error from the departure between the observation and the background (expel: Hollingsworth, A., and P. Lonnberg, 1986), or using a surrogate quantity whose error statistics can be a good approximation of the unknown background errors (such as NMC). More recent approaches rely on ensembles of analyses. I suggest to provide here more background information on existing approaches to model the BEC including their advantages and limitations.

**Response** We focused on the BEC modeling method used in this study, so that we decided to keep the paragraph unchanged here.

**Comment** line 69 : "Among the greatest sources of errors in CTM simulations (e.g., Elbern et al., 2000; Wu et al., 2008) are the uncertainties of emission inventories". This sentence should be moved to the paragraph listing the sources of uncertainties affecting CTM simulations.

**Response** We moved this sentence into the paragraph discussing the source of uncertainties affecting CTM simulations. Please, see pp. 2:52–pp. 2:53.

**Comment** Paragraph on methods to account for uncertainties in emission inventories: Not enough background is given on this central aspect of this paper. There are studies that have attempted to include the emission fluxes in the control vector and estimate them using ensemble data assimilation approach.

**Response** We have added the sentence: "On the other hand, another recent study developed an ensemble data assimilation approach in order to constrain not only initial conditions of carbon monoxide (CO), but also surface CO emission fluxes (i.e., dual assimilation-inversion) (Barré et al., 2019). It was noted in this study that the error distributions in surface emissions play an important role in the performance of CO predictions.". Please, see pp. 3:78–pp. 3:81.

**Comment** need to clarify that $PM_{2.5}$ is the output variable targeted in this work. What is the rationale for choosing PM2.5 instead of AOD?

**Response** Again, main focus (objective) of this study was/is to improve short-term (24-hour) $PM_{2.5}$ predictions in South Korea. This was the main reason that we chose $PM_{2.5}$ as control variable, instead of AOD. We already mentioned the availability issue of AOD in East Asia. Moreover, if we selected the satellite-retrieved AOD as control variable for the purpose of short-term predictions in South Korea, we definitely need one more key technique to convert AOD into surface $PM_{2.5}$. Although this technique has been developed at authors' lab using several machine learning-based techniques, it will bring more uncertainties into our framework. All the reasons mentioned above are why we chose $PM_{2.5}$ as control variable, not AOD in this study. In order to clarify this, we have added/modified the sentences. Please, refer to the paragraph at pp. 3:82–pp. 3:90.

**Comment** The structure of Section 2 needs substantial revision.
Section 2.1 includes several aspects that should be included in separate subsections, I suggest the following structure:
a/Study site and observations: the second paragraph of Section 2.1 concerns the 'description of the study it is not clear how the observations used for validation and data assimilation were selected. What is the vertical footprint of the measured PM2.5? How does it compare with the modelled value?

**Response** We significantly modified and re-structured the paragraphs. please, refer to the pp. pp. 9:262–pp. 9:268 and pp. 11:333–pp. 11:340. We did not use observations having vertical footprint in this study. Such types of observations have not been available in East Asia.

**Comment** b1/model description: a short description of the CMAQ CTM is missing. Providing the version of the aerosol and chemistry module is not enough, key references are missing. I suggest to give the main characteristics of aerosol and chemistry schemes (e.g. number of species and reactions for the chemistry, list of aerosol species for the aerosol scheme) along the main characteristics of the atmospheric transport model:(which type of advection scheme is used )

b2/model configuration: it should address time and spatial resolution, coupling between WRF and CMAQ, temporal period, location, output variables.

**Response** We added a new table providing information on the options of WRF and CMAQ model simulations. Table will provide some details on both meteorological model and CTM simulation. Please, check out pp. 4:121–pp. 4:129 as well as Tables S1 and S2.

**Comment** Emission datasets: Can you justify why these two data sets have been selected for this work .

**Response** These two emission data sets were the most reliable in East Asia. Please, check out the details at pp. 5:161–pp. 6:167.

**Comment** Data assimilation and BEC modelling: Since BEC modelling is a central aspect of the methodology, a dedicated section should explain how it is parametrized and how the NMC method is used to estimate the BEC parameters in this work.

**Response** We seriously modified the paragraphs. Please, see pp. 7:215–pp. 7:223.

**Comment** Experiment design: This section should include the statements given from line 178 to 199. A table summarizing all the experiments/simulation could be helpful. The various cases of implementation of the NMC method need some clarifications.

**Response** We have modified Fig. 4. and added a table summarizing the experiment design. Please, check out Fig. 4. and Appendix A.

**Comment** Validation methodology: a clear definition of PM2.5 is missing: what is the vertical footprint of $PM_{2.5}$? What are the differences between the modelled and the observed $PM_{2.5}$?

**Response** Please, refer to pp. 2:38. As mentioned previously, we did not use observations related to the vertical footprint of $PM_{2.5}$ (such data was not available in East Asia!). Simulated $PM_{2.5}$ was calculated using simulated aerosol species and aerosol size fraction, and observed $PM_{2.5}$ was measured values. Please, see pp. 6:191–pp. 7:197 for the definition of modeled $PM_{2.5}$ used in this study.

**Comment** Some aspects of the methodology are not clear or not accurate enough
Section 2.1
line 123: which conserving method? please give a reference

**Response** We have used a program named "Spatial Allocator" for flux-conserving interpolation. The "Spatial Allocator" distributed by Community Modeling and Analysis System (CMAS) has been used for emission manipulations. Please, refer to the paragraphs at pp. 5:137–pp. 5:138.

**Comment** line 125-130: this belongs to Results and not to Methodology section. "The differences in South Korea are relatively small, except for CO in the MIX emission inventory.": Are you talking about the differences between the two databases? I do not understand "except for CO in the MIX inventory"

**Response** We have modified this paragraph. Please, see pp. 5:154–pp. 5:156.

**Comment** line 130-136 on the use of MEGAN. Why are you using LAI from MODIS and GVF from VIIRS? Are these variables required to drive MEGAN? There is a possible inconsistency between LAI from MODIS and GVF from VIIRS? Can you comment on it?

**Response** It would be the best option to use both LAI and GVF retrieved from the same satellite. However, both GVF from MODIS and LAI from VIIRS were not available, when our study began. MEGAN v3 requires LAIv which is the ratio of LAI to GVF. We used 8-day averaged LAI and GVF, so that we assumed that the inconsistency may not be a major issue.

**Comment** the description of the cost function (l14-151) is a bit confusing. x is the control vector. X and xb contain the same variables (both are of the same size). x is the analysis and xb is the background. Are you also assimilating other variables which drive the chemistry or the transport model ? line 166 redundancy with Introduction line 172, not accurate definition of S: S represents the background error and its diagonal components are the standard deviation of the error of the background. What are the differences between the measured and the simulated $PM_{2.5}$?

**Response** We did not assimilate other variables. We have removed the redundant sentence you pointed out. We have modified the definition of S. Please, see pp. 6:212–pp. 6:213. The difference between the measured and the simulated $PM_{2.5}$ was mentioned previously.

**Comment** line 211, eq 4: How a and b values have been chosen? line 213: replace 'second criterion' by 'Eq 4 criterion' line 213-215: this belong to the data assimilation section/BEC description. Some parts of the methodology are lacking such as the selection of observations for data assimilation versus validation.

**Response** We have added the references related to those values. Please, see pp. 9:260. We replace 'second criterion' by 'criterion of Eq 4'. We have also relocated the sentence describing BEC. Please, see pp. 6:187–pp. 6:189. For the selection of observations, please refer to pp. 9:262–pp. 9:268.

**Comment** Section 3 should be dedicated to the presentation of the results. A separate section should address the discussions points. I shortly review the results but further review of them should be done if the manuscript is considered for publication.

- line 221-222: "To estimate the influence of the two : : :" : this belongs to the previous section

**Response** In order to further clarify this point, we have changed the paragraph. Please, check out pp. 9:270–pp. 9:275.

**Comment** why incorporating emission uncertainties in BEC should influence the vertical distribution for $PM_{2.5}$?

**Response** Uncertainties in emissions are certainly related to the uncertainty in surface $PM_{2.5}$. However, because atmospheric species are transported vertically due to turbulent convection processes, the vertical distributions of $PM_{2.5}$ can also be influenced by the uncertainties in emissions.

**Comment** l248: "In the DA process, the horizontal length scale determines $PM_{2.5}$ increases in the horizontal spread of analysis" I do not understand this statement. The horizontal length scale refers to the horizontal correlation of $PM_{2.5}$

**Response** We have modified the original sentence like following: "the horizontal length scale determines the horizontal spread of $PM_{2.5}$ increments around observation locations (Descombe et al., 2016)".

**Comment** line 258: "The characteristics of the vertical and horizontal length scales, however, have not been fully explained in this study, thus requiring future": The authors should further discuss this aspect and provide possible explanations.

**Response** We intended to explain that further studies are necessary and that the studies should be carried out in the future. Regarding this point, please, check out pp. 12:391–pp. 12:399.

**Comment** Section 3.2 first Paragraph: This belongs to methodology and should be described in the experiment design section.

**Response** We have moved this paragraph into the section of experiment design.

**Comment** The last two paragraph should be developed in a separate discussion Section. Part of it should also be used as background information in Introduction. I can see also some redundant ideas from the Kumar et al, 2019 paper.

**Response** We have changed the section name from "Conclusion" to "Summary and Conclusions". Here, we intended to mention some limitations of current work.

**Comment** Lack of references in several part of the papers

**Response** We have tried to provide more relevant references in the revised papers.

**Comment** The use of a large number of acronyms makes the reading somehow very difficult.

**Response** We thought that we have used the acronyms commonly used in the atmospheric studies. To avoid the difficulty, we decided to add a list of acronyms related to the experiment design. Please, refer to Appendix A.

**Comment** Result description needs to be improved, some sentences are confusing.

the style is frequently not appropriate with a lot of uncertain and long sentences: for example "We found that the new approach exhibited a tendency to generate substantially increased standard deviations" , a tendency to generate : : : , I suggest using more direct sentences.

**Response** We have tried to modify several indirect sentences throughout the manuscript. Please, check out pp. 12:384 – pp. 12:385.

---

## Author Comment (AC4) · 18 Nov 2020

**Response to reviewer 4**

We appreciate reviewer's thoughtful comments and suggestions, which are greatly helpful for us to improve our manuscript. The manuscript has been revised to accommodate the reviewer's comments and suggestions.

**General comment** Given the structure of the 3DVAR cost function, any increase in BEC will lead to the analysis being closer to the observations, and further away from the model. Therefore, the reported better fit of the analysis with PM2.5 observations is consistent. However, there is the danger of statistical overfitting especially when the observation errors are potentially chosen to be too small. To convincingly demonstrate the improvement in the analysis requires a cross-validation approach: The randomly selected subset of the observations which is used for the evaluation should not be assimilated. I strongly recommend to carry out such a test to demonstrate the impact of the new BEC, especially the for the length scales, in a scientifically clean way.

**Response** To answer the review's comments, we carried out an additional analysis experiment with 20% of independent observations which were randomly taken out of the observations in the DA process and were used only for comparison purpose. Please, refer to pp. 9:262–pp. 9:268 and pp. 11:333–pp. 11:340 in the revised manuscript.

**General Comment** Besides the BEC, the choice of the observation error standard deviation influences the match between the assimilated observations and the analysis. The authors should therefore provide more details how the representativeness error (i.e. of the station observations for the corresponding model grid box) of the observation has been considered. It seems that the presented approach only accounts for the instrument error but not the representativeness error. It would be interesting to see if decreasing observation error SD has a similar influence on the analysis than increasing BEC. BEC and observations error need to be discussed in together as their relative differences determines the match of the observations with the analysis.

**Response** We totally agree with your comment that the balance between observation error (including the representativeness error) and background error determines the performances of data assimilation and short-term $PM_{2.5}$ predictions. However, we did not consider the

representativeness error (also known as "sub-grid problem") in current study. As far as we understand, several groups (including authors' lab) have tried to evaluate these errors using an artificial intelligence technique. We have added a brief discussion of this point. Please, refer to pp. 13:400 – pp. 13:405.

**General comment** While it is acknowledged that the uncertainty of the emissions should be considered in the background error statistics, this quantification is not easy. Comparing two different emission data set, as done by the authors, will be dominated by the biases between the two data sets. The paper should provide more evidence that the documented in increase in BEC is a consistent estimate of the (unbiased) uncertainty of the emissions. The reader wants to know what the resulting uncertainty estimate of the emissions is.

**Response** Again, we agree with reviewer's opinion. The biases between the two emission inventories dominate the comparisons of different emissions. Therefore, we removed the mean of perturbations for estimating BEC using GEN-BE v2.0. It is impossible to know the uncertainties of all chemical species in emission inventories. Please, note that the differences in the two emissions were not artificially created, but they were based on two fully independent emissions that had been established with two different techniques and statistical factors (such as emission factors and profiles, activity data, pollution control efficiency, regulation penetration, and economic growth rates) in East Asia. Given such difficulty in estimating the uncertainty in single emission inventory, we decided to use two independent emissions in our study to take into account the uncertainty in emissions in East Asia. Relevant discussions are introduced in our revised manuscript, too. Please, check out pp. 5:161 – pp. 5:167.

**Comment** The title suggests that the reader will get an information about the uncertainty of the emissions. This seems not the case.

**Response** Readers will get some information on "the impacts of uncertainty in emissions on data assimilation and short-term prediction in East Asia". We have changed the title as: "An investigation into the impacts of uncertainties in emissions on aerosol data assimilation and short-term $PM_{2.5}$ predictions using CMAQ v5.2.1".

**Comment** l 28: Please provide a quantification of the increase in the BE SD by taking the emission uncertainty into account.

**Response** We provided the quantification of the BEC SD. Please, refer to pp. 1:23–pp. 1:24.

**Comment** l 53 please discuss here also the representativeness error with respect to your model resolution

**Response** As mentioned above, we did not consider the representativeness error, but we have added a brief discussion on that errors. Please, see pp. 13:400 – pp. 13:405.

**Comment** l 130 Please comment if any additional temporal profiles (diurnal cycle, weekly cycle etc) were applied to the emissions during the simulation. The temporal variability might be a large source for the uncertainty of the emissions.

**Response** Good points! But, there is no detailed information available in our domain, as far as we know (although we used a temporal profile of the CREATE in our study!). This is obviously a topic we should work on more in the future. We have made some comments on the temporal profiles of emission inventories. Please, see pp. 5:144–pp. 5:145.

**Comment** l 149 Theoretically speaking, the error in H is the representativeness error

**Response** We have added a sentence into pp. 6:187.

**Comment** l 154 Please explain if the PM components are also modified by the DA or only the diagnostic PM2.5 field.

**Response** Diagnostic $PM_{2.5}$ fields estimated by Eq. 1 was modified by the DA. Regarding this point, please see pp. 7:198–pp. 7:206.

**Comment** l 185 Please provide more details here. PM2.5 simulations using the same emissions

but different meteorological fields (i.e from different forecast lead times) can expected to be unbiased (i.e. inly a random error). However, this will not be the case if different emission data are used. How does the biases in the emissions turn into increased SD of the background errors. Did you remove these biases in the calculation of the variances and covariances as expected from the definitions of these statistical parameters.

**Response**: Once GEN_BE v2.0 used for estimating BEC starts to calculate the perturbations, it removes the mean of perturbations (i.e., "Stage 1" in the program). After that, the calculations of the variances and covariances start. Not direct bias correction was made on emissions because we selected $PM_{2.5}$ as the control variable. The perturbations are from simulated $PM_{2.5}$ variations caused by the changes in meteorology and/or emission inventories.

**Comment** l 214 I am not sure I understand this formulae. The term seems completely dominated by the 50 microgram/m3 constant value. Are you saying the SD of the observation errors is more or less 50 micorgram/m3 all the time? That would be quite a lot. Please compare the observation error SD against the SD of the background error (2, 4 or 8 microgram, see Fig 5)

**Response** We have corrected the value which is originally 1.5 $\mu g$ $m^{-3}$ from Schwartz et al (2012). The value given in the original manuscript was just a typo in the previous manuscript, and we used the value of 1.5 in our real work. Thank you for your comment and correction!

**Comment** l 224 Please express these numbers also in percent w.r.t to typical PM2.5 values.

**Response** We could not find the line you made a comment on. Could you provide the line? We will then correct the numbers.

**Comment** l 278 Please see my general comment. I think it is necessary to use independent (i.e. not used in the assimilation) observations to estimate the quality of the analysis.

**Response** As we answer in **General comment**, we carried out additional comparison of DA results with 20% of independent observations, which were randomly taken out of the

observations and were then used for comparison purpose. Please, refer to pp. 9:262–pp. 9:268 and pp. 11:333–pp. 11:340 in the revised manuscript.

**Comment** L 298 Please provide more detail, how the analysis for diagnostic PM25 (formulae 2) is converted back into the prognostic aerosol variables.

**Response** We have added a sentence into pp. 7:198–pp. 7:206.

**Comment** l 325 Please provide a quantitative information about the assumed or inferred estimate of emission uncertainty.

**Response** It is difficult to find a way to quantify the emission uncertainty. As far as we are aware, it is almost impossible job to quantify the uncertainty of a bottom-up emission inventory. Again, given the difficulty in estimating the uncertainty in single emission inventory, we decided to use two independent emissions in our study to take into account the uncertainty in emissions in East Asia. Relevant discussions are introduced in our revised manuscript. Please, refer to pp. 5:161–pp. 6:167.

**Comment** l 328 A proper cross validation is needed to avoid overfitting (see general comment).

**Response** Again, please refer to added/modified paragraphs (pp. 9:262–pp. 9:268 and pp. 11:333–pp. 11:340).

**Comment** l 345 Please compare the SD of your BEC with the one derived with ensemble methods quoted in the literature.

**Response** The references you mentioned are NWP studies. The SDs in those literatures are all related to the meteorological variables. We think that the ensemble method used in recent NWP studies may be applicable to chemical DA. That is going to be our future topic to investigate.